# HOI-Dyn: Learning Interaction Dynamics for Human-Object Motion Diffusion

**Lin Wu**
James Watt School of Engineering
University of Glasgow
l.wu.1@research.gla.ac.uk

**Zhixiang Chen**
School of Computer Science
University of Sheffield
zhixiang.chen@sheffield.ac.uk

**Jianglin Lan**[†]
James Watt School of Engineering
University of Glasgow
jianglin.lan@glasgow.ac.uk

## Abstract

Generating realistic 3D human-object interactions (HOIs) remains a challenging task due to the difficulty of modeling detailed interaction dynamics. Existing methods treat human and object motions independently, resulting in physically implausible and causally inconsistent behaviors. In this work, we present HOI-Dyn, a novel framework that formulates HOI generation as a driver-responder system, where human actions drive object responses. At the core of our method is a lightweight transformer-based interaction dynamics model that explicitly predicts how objects should react to human motion. To further enforce consistency, we introduce a residual-based dynamics loss that mitigates the impact of dynamics prediction errors and prevents misleading optimization signals. The dynamics model is used only during training, preserving inference efficiency. Through extensive qualitative and quantitative experiments, we demonstrate that our approach not only enhances the quality of HOI generation but also establishes a feasible metric for evaluating the quality of generated interactions. Project website: https://wulin97.github.io/hoi-dyn

## 1   Introduction

Synthesizing complex and realistic 3D human-object interactions (HOIs) is essential for progress in VR/AR, computer animation, and robotics [1–4], yet remains a significant challenge. Compared to human motion generation—whether for single or multiple people [5, 6]—HOI is significantly more difficult. Human motion generation typically involves relatively free movement, making it easier to generate plausible sequences [7, 8]. However, HOI requires capturing intricate interaction dynamics, such as stable contact, forces, and action-response relationships. Simply applying human motion generation frameworks to HOI often produces independent motion of the human and object, leading to physically unrealistic and causally inconsistent behaviors.

Previous works on HOI generation mainly focused on interactions between humans and static objects or scenes [9–12], such as sitting on a sofa. Recent advances have shifted toward controllable synthesis of dynamic HOI, where both the human and object move synchronously [13, 14]. For example, given an initial state and a textual instruction, the system can generate a motion sequence where a human

---

[†]Corresponding author

39th Conference on Neural Information Processing Systems (NeurIPS 2025).

picks up an object, such as a bench, and places it elsewhere. This approach enables more flexible motion generation and a wide range of applications.

However, existing methods often fail to capture the core interaction dynamics between humans and objects. These approaches typically focus on modeling either object affordances or contact points [1, 4, 15], or simply integrating human and object motions through diffusion-based models [2, 13]. However, they do not fully address how objects should respond to human actions, often leading to physical and causal inconsistencies.

In this work, we propose a new perspective: *framing HOI generation as a driver-responder system [16], where human actions serve as the driver and objects respond accordingly. At the heart of this approach is the modeling of interaction dynamics, which describes how objects should naturally react to human motions.* This view offers several advantages:

- **Contact is implicitly governed by the dynamics**—*no need to explicitly model it. If there is no contact, there is no response; if contact occurs, the object's response is naturally determined by the interaction dynamics.*
- **Object motion is not independent**—*each step of their movement is driven by the human's actions and controlled through specific instructions or context, ensuring a coherent and physically plausible interaction.*

Building on this perspective, we design a new HOI generation framework that explicitly incorporates interaction dynamics into the motion synthesis process, yielding state-of-the-art performance on challenging HOI benchmarks and offering a physically grounded solution to HOI generation. Specifically, our contributions are as follows:

- We introduce a novel **driver–responder formulation** for HOI generation from a synchronized control perspective, modeling the causal dependencies between human actions and object responses in a dynamic and physically consistent manner.
- We propose a lightweight transformer-based **interaction dynamics model** that answers how objects should react dynamically to human actions, taking into account the context of human motion and specific contact situations.
- We introduce a **residual-based interaction dynamics loss** that serves HOI motion diffusion, compensating for prediction noise in the dynamics model. This loss helps prevent misleading optimization gradients and ensures the focus remains on core generative inconsistencies, thereby improving the quality of the generated motion sequences.
- We demonstrate the effectiveness of our approach through extensive qualitative and quantitative experiments, highlighting that the proposed model not only improves HOI generation but also serves as a reliable evaluation metric for assessing the quality of generated interactions.

## 2 Related Work

### 2.1 Object-Guided HOI Generation

A subset of HOI generation methods leverages object trajectories or waypoints to guide human motion. OMOMO [1] takes a full sequence of object states as input and generates the corresponding human poses, whereas CHOIS [13] and Wu et al. [2] rely on sparse object waypoints (e.g., roughly one waypoint every 30 frames), leaving the object's detailed responses to be implicitly determined by the model. These methods typically add auxiliary supervision or constraints to encourage plausible human-object contact. However, such guidance mainly steers the diffusion process toward ensuring that contact occurs, rather than modeling the underlying interaction itself. While this strategy can yield globally coherent sequences, it remains at a high level: the model is not trained to capture how objects physically react to human actions in a fine-grained and causally consistent manner, leaving the central challenge of realistic HOI underexplored.

### 2.2 Joint Human-Object Motion Generation

Another line of work generates HOIs jointly without conditioning on future object states. HOI-Diff [17] relies on affordance prediction and estimated contact points to guide interactions, while

CG-HOI [15] leverages contact fields on the human mesh as strong priors. THOR [14] models relational cues for coordinated motions, HIMO [18] employs a dual-branch conditional diffusion with a mutual interaction module for cross-modal fusion, and ChainHOI [19] adopts a spatiotemporal graph architecture with a kinematics-aware module to capture joint- and chain-level dependencies.

Despite these advances, common challenges remain. First, predicting accurate contact points or modeling interactions precisely at contact regions is inherently difficult. Second, although high-level constraints or interaction modules encourage consistency, they often fail to capture the causal and fine-grained dynamics of object responses to human actions. As a result, object behaviors may appear temporally inconsistent or physically implausible, even when global plausibility is achieved. This highlights a missing perspective: explicitly treating human actions as the driver and object motions as their causal responses. Such a formulation is crucial for achieving physically realistic HOI generation.

### 2.3 Driver-Responder Synchronization

Driver-Responder Synchronization, also referred to as master-slave synchronization, has been widely observed in coupled systems across biological and physical domains [20–22]. In these systems, controllers achieve synchronization through adaptive and feedback strategies [23, 24]. Drawing inspiration from these systems, we conceptualize HOI generation as a *Driver-Responder System*, where human actions serve as the driver, controlling the object's response. Unlike prior methods that treat human and object motions independently [13, 25, 26], this perspective captures the causal relationship between human movements and object reactions, ensuring more coherent and physically consistent interactions. By introducing an internal control mechanism, the Driver-Responder formulation provides a principled framework for generating HOIs with refined temporal coherence and realistic object dynamics, addressing the limitations of previous high-level constrained approaches.

## 3 Methodology

Our goal is to synthesize synchronized HOIs while maintaining internal causal consistency, by leveraging controllable signals such as textual descriptions and object geometry. We propose the *HOI-Dyn* framework (see Fig. 1) that explicitly models interaction dynamics, enabling the generation of more plausible and coherent motion sequences. The framework consists of two key components: **Motion Diffusion** and **Interaction Dynamics**. The Motion Diffusion component, based on a Transformer-based conditional diffusion model, jointly encodes the human, object, and interaction context into a unified representation. The Interaction Dynamics component provides auxiliary supervision to reinforce fine-grained causal consistency during motion generation.

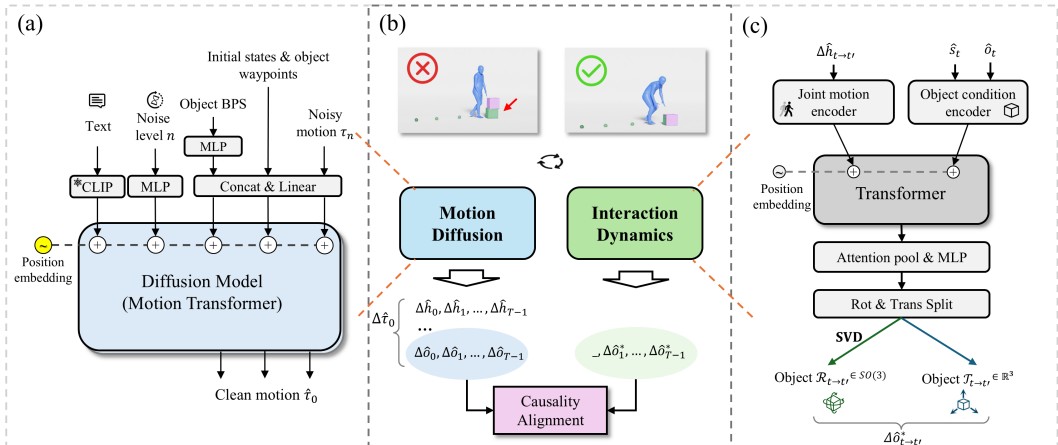

Figure 1: **Overview of the proposed HOI-Dyn framework.** (a) *Conditional Motion Diffusion* synthesizes human-object interactions $\hat{\tau}_0 = \{\hat{H}, \hat{O}, \hat{X}\}$ using a Transformer-based diffusion model, where $\hat{H} := \{\hat{h}_t\}_{t=0}^{T-1}$ and $\hat{O} := \{\hat{o}_t\}_{t=0}^{T-1}$. (b) The full framework integrates motion generation with interaction dynamics supervision. (c) *Interaction Dynamics* models object responses $\Delta\hat{o}_t^*$ based on human relative motion $\Delta\hat{h}_t$, object pose $\hat{o}_t$, and interaction context $\hat{s}_t$.

## 3.1 Interaction Dynamics

In diffusion-based motion generation, the output typically reflects clean motion rather than noise. The denoising process allows the model to internalize human-specific dynamics, resulting in physically plausible free movement. In contrast, object motion is externally driven and inherently constrained by physical laws—it cannot occur autonomously. To reflect this asymmetry, we decouple their roles as follows: the human follows internally guided dynamics, while the object responds to external control. This interaction is formalized as

$$\textbf{Driver (Human)}: \begin{cases} h^{(t+1)} = h^{(t)} + \Delta t \cdot F_h(h^{(t)}) \\ y_h^{(t)} = g_h(h^{(t)}) \end{cases},$$

$$\textbf{Responder (Object)}: \begin{cases} o^{(t+1)} = o^{(t)} + \Delta t \cdot F_o(o^{(t)}, s^{(t)}, u^{(t)}) \\ y_o^{(t)} = g_o(o^{(t)}) \end{cases}, \tag{1}$$

where $h^{(t)}$ and $o^{(t)}$ denote the latent states of the human and the object at time step $t$, respectively. The functions $F_h(\cdot)$ and $F_o(\cdot)$ describe their respective internal dynamics, while $g_h$ and $g_o$ project these latent states to the observable outputs $y_h^{(t)}$ and $y_o^{(t)}$. $\Delta t$ is the sampling time step. The control signal $u^{(t)}$ is determined based on the error feedback $e(y_h^{(t)}, y_o^{(t)})$, which reflects the discrepancy between the human's intent and the object's behavior. The term $s^{(t)}$ denotes the interaction context, which includes factors such as contact state, object geometry, and other environment- or task-specific conditions influencing the object's motion.

In practice, we observe that existing diffusion models often fail to generate causally consistent interactions due to the absence of precise control signal $u^{(t)}$, which is essential for aligning the object's motion with human intent [13, 14]. To address this limitation, we propose the interaction dynamics as a supervisory signal to implicitly optimize it in the diffusion process. Based on (1), we can derive the object's relative motion as

$$\Delta o^{(t)} = o^{(t+1)} - o^{(t)} = \Delta t \cdot F_o(o^{(t)}, s^{(t)}, u^{(t)}) \approx \mathcal{D}(s^{(t)}, o^{(t)}, \Delta h^{(t)}; \theta_{\mathcal{D}}), \tag{2}$$

where $\Delta h^{(t)} = h^{(t+1)} - h^{(t)}$ denotes the human's relative motion, and $\mathcal{D}(\cdot)$ is a learnable function parameterized by $\theta_{\mathcal{D}}$. This formulation emphasizes that object dynamics are governed not only by their internal state but also by human-induced interactions and the context.

To enhance sensitivity to varying interaction magnitudes, we extend the HOI prediction horizon from 1 to $k$, with $k$ being selected randomly from $[1, K]$. As shown in Fig. 1(c), the model takes as input the current object state $o^{(t)}$, interaction context $s^{(t)}$, and the cumulative human motion over the horizon $\Delta h_{t \to t+k} = h^{(t+k)} - h^{(t)}$, and predicts the corresponding object motion:

$$\Delta o_{t \to t+k}^* \approx \mathcal{D}(s^{(t)}, o^{(t)}, \Delta h_{t \to t+k}; \theta_{\mathcal{D}}). \tag{3}$$

The above predicted motion is further represented as a rigid-body transformation comprising a rotation $\hat{\mathcal{R}}^{(t \to t+k)} \in \mathrm{SO}(3)$ and translation $\hat{\mathcal{T}}^{(t \to t+k)} \in \mathbb{R}^3$, i.e., $\Delta o_=^*[\hat{\mathcal{R}}|\hat{\mathcal{T}}]$, applied to a set of object's points $\mathcal{P}^{(t)}$ to yield the predicted future configuration as follows:

$$\hat{\mathcal{P}}^{(t+k)} = \hat{\mathcal{R}}^{(t \to t+k)} \mathcal{P}^{(t)} + \hat{\mathcal{T}}^{(t \to t+k)}. \tag{4}$$

To ensure that $\hat{\mathcal{R}}^{(t \to t+k)}$ is a valid rotation matrix, we apply singular value decomposition (SVD)-based projection to the raw network output $\tilde{\mathcal{R}} \in \mathbb{R}^{3 \times 3}$, such that $\tilde{\mathcal{R}} = U\Sigma V^\top$ and $\hat{\mathcal{R}} = UV^\top$. We then define the following cost for object dynamics to quantify the error between the transformed keypoints:

$$\Phi(\Delta o_{t \to t+k}, \Delta o_{t \to t+k}^*) = \|\mathcal{P}^{(t+k)} - \hat{\mathcal{P}}^{(t+k)}\|_1. \tag{5}$$

The overall loss function is defined as the expected value over time steps $t$ and $k$ sampled from $\mathcal{U}(1, K)$ as follows:

$$\mathcal{L} = \mathbb{E}_{t, k \sim \mathcal{U}(1,K)} \left[ \frac{1}{k} \cdot \Phi(\Delta o_{t \to t+k}, \Delta o_{t \to t+k}^*) \right]. \tag{6}$$

This loss function guides the model to capture both the motion magnitude and the causal structure of interactions, yielding more realistic and consistent object motion in HOI.

## 3.2 Conditional Motion Diffusion

Having introduced the interaction dynamics, we now proceed to the motion generation stage. Here, we propose a conditional diffusion model that provides internal forces to synthesize temporally aligned motions for both the human and the object.

The joint human-object trajectory is denoted as $\tau = \{H, O, X\}$, with the human motion $H$, object motion $O$, and interaction context $X$ (e.g., hand-object and foot-object contact annotations). The context provides coarse guidance for synthesizing plausible interactions. We represent human motion using the SMPL-X parametric model [27], and represent each frame of the object motion using 3D translation and relative rotation. In the condition representation $\mathbf{c}$, as shown in Fig. 1(a), we follow the CHOIS framework [13] to integrate contextual cues such as text prompts and the Basis Point Set (BPS) [28] of the object. More details can be found in Appendix B.

The complete conditional HOI Diffusion comprises both forward and reverse processes [29]. The forward process is modeled as a Markov chain over $N$ steps as follows:

$$q(\tau_{1:N}|\tau_0) = \prod_{n=1}^{N} q(\tau_n|\tau_{n-1}), \quad q(\tau_n|\tau_{n-1}) = \mathcal{N}(\tau_n; \sqrt{1-\beta_n}\,\tau_{n-1}, \beta_n\mathbf{I}), \tag{7}$$

where $\beta_n$ is a predefined noise schedule.

The reverse process is defined as

$$p_\theta(\tau_{n-1}|\tau_n, \mathbf{c}) = \mathcal{N}(\tau_{n-1}; \mu_\theta(\tau_n, n, \mathbf{c}), \Sigma_n), \tag{8}$$

where $\Sigma_n$ is a fixed variance, and $\mu_\theta = \frac{\sqrt{\alpha_n}(1-\bar{\alpha}_{n-1})}{1-\bar{\alpha}_n} \cdot \tau_n + \frac{\sqrt{\bar{\alpha}_{n-1}}\beta_n}{1-\bar{\alpha}_n} \cdot \hat{\tau}_0$, with $\bar{\alpha}_n = \prod_{i=1}^{n} \alpha_i$ and $\alpha_n = 1 - \beta_n$.

The reverse process (8) is modeled by a neural network $\theta_{\mathcal{G}}$ with the training loss:

$$\mathcal{L}_{\text{hoi}} = \mathbb{E}_{\tau_0, n}\left[\|\hat{\tau}_0(\tau_n, n, \mathbf{c}; \theta_{\mathcal{G}}) - \tau_0\|_1\right], \tag{9}$$

which minimizes the reconstruction error between the predicted and ground-truth trajectories.

Although the standard loss function (9) produces plausible motion sequences, physical inconsistencies persist, particularly in object trajectories. To address this, we introduce an auxiliary **Interaction Dynamics Loss** based on a shared dynamics model $\mathcal{D}$. This loss penalizes discrepancies in object motion between generated and ground-truth sequences, enforcing a **causal alignment** between human actions and object responses, as follows:

$$\mathcal{L}_{\text{dyn}} = \mathbb{E}_t\left[\|\Phi(\Delta\hat{o}_t^*, \Delta\hat{o}_t) - \Phi(\Delta o_t^*, \Delta o_t)\|_1\right], \tag{10}$$

where $\{\Delta\hat{o}_t^*\}_{t=1}^{T-1} = \mathcal{D}(\Delta\hat{\tau}_0)$ and $\{\Delta o_t^*\}_{t=1}^{T-1} = \mathcal{D}(\Delta\tau_0)$ are the object motions predicted by $\mathcal{D}$ given the relative motion from the generated and ground-truth HOI $\hat{\tau}_0$ and $\tau_0$, respectively. The function $\Phi(\cdot, \cdot)$ computes the motion errors as described in (5). The per-frame residuals in $\mathcal{L}_{\text{dyn}}$ are

$$\delta_{D,\text{gen}} = \Phi(\Delta\hat{o}_t^*, \Delta\hat{o}_t), \quad \delta_{D,\text{gt}} = \Phi(\Delta o_t^*, \Delta o_t). \tag{11}$$

Ideally, if $\mathcal{D}$ perfectly captures the interaction dynamics, then $\delta_{D,\text{gt}} = 0$ and the loss reverts to the direct supervision of $\delta_{D,\text{gen}}$. However, in practice, prediction errors are unavoidable due to imperfections in $\mathcal{D}$ and inaccuracies in the training data.

To mitigate the influence of these errors, we assume that $\mathcal{D}$ is locally smooth and time-homogeneous—its bias depends on the state but not on time $t$. This assumption is reasonable because $\mathcal{D}$ predicts object dynamics from current states without explicit time dependence, and local smoothness reflects the continuity of physical motion (see Appendix E.2 for a detailed discussion of these assumptions). Under this assumption, when the generative and ground-truth trajectories exhibit similar state distributions, the error residuals tend to cancel out in expectation:

$$\mathbb{E}_t[\delta_{D,\text{gen}} - \delta_{D,\text{gt}}] \approx 0. \tag{12}$$

This residual formulation provides a robust supervisory signal: even if $\mathcal{D}$ is imperfect, its bias is removed through subtraction, allowing the learning algorithm to focus on the true inconsistencies in the generated motion. For a more detailed discussion, see Appendix C. With this loss in place, each step of the generation process is guided to yield a more accurate control signal $u_t$, resulting in physically grounded and temporally coherent HOI sequences. We therefore extend the standard HOI training objective $\mathcal{L}_{\text{hoi}}$ (9) by incorporating both $\mathcal{L}_{\text{dyn}}$ (10) and an object-level reconstruction loss $\mathcal{L}_{\text{obj}} = \mathbb{E}_t[\Phi(o_t, \hat{o}_t)]$. All these losses are equally weighted to simplify the overall training process and avoid the need for complex hyperparameter tuning.

# 4 Experiments

**Dataset.** We train and evaluate HOI generation using two datasets: (i) *FullBodyManipulation*, which provides 10 hours of high-quality paired object and human motion data involving 15 different objects [1]. We apply the OMOMO partitioning to this dataset: 15 subjects for training and 2 for testing; and (ii) *3D-FUTURE*, which consists of 3D models of various furniture items [30]. To assess the generalization capability, we select 17 objects and pair them with motion sequences from the *FullBodyManipulation* testing set.

**Metrics.** We evaluate HOI from four aspects, as defined in CHOIS [13]: (i) *Condition Matching*, which measures the alignment of the predicted object trajectory with input waypoints using Euclidean distance at the start, end, and intermediate points (cm). (ii) *Human Motion Quality*, which is evaluated with the foot sliding score (FS), foot height $H_{\text{feet}}$, and Fréchet Inception Distance (FID) [31] that captures distributional differences between the generated and real motions. (iii) *Interaction Quality:* Based on the frame-wise contact labels, we compute the contact percentage $C_\%$, F1 score $C_{F_1}$, and hand-object penetration score $P_{\text{hand}}$ to assess physical plausibility. (iv) *Ground Truth Difference*, which includes the mean per-joint position error (MPJPE), root translation error $T_{\text{root}}$, object position error $T_{\text{obj}}$, and orientation error $R_{\text{obj}}$.

**Implementation Details.** We train the interaction dynamics using *FullBodyManipulation* with $K = 2$ as the maximum prediction horizon. The network has 0.5M parameters. We use the Adam optimizer [32] with a learning rate of $1 \times 10^{-3}$, adjusted via *CosineAnnealingWarmRestarts* [33] for 150 epochs with a batch size of 32. The model is then transferred to the HOI motion diffusion task, where fine-grained driver-responder relationships are encouraged during denoising, as in (10). Training is done from scratch with a learning rate of $1 \times 10^{-4}$, batch size 32, and 100,000 steps. All experiments are conducted on a single NVIDIA RTX A4500 GPU, with total training time around 10 hours.

**Baselines.** We compare our method HOI-Dyn with the state-of-the-art (SOTA) approach, CHOIS [13], whose results are reproduced under identical training conditions, including the same number of training steps. We also consider other baselines including the adapted versions of Inter-Diff [34], MDM [35], and OMOMO [1], whose results are borrowed from [13] for comparison. More specifically, InterDiff is modified to accept additional inputs, including text and sparse waypoints. MDM is updated to incorporate our object geometry representation and sparse waypoints, extending it to predict object motion. We introduce three variants of OMOMO: (i) Pred-OMOMO, which combines our object motion module with OMOMO; (ii) GT-OMOMO, which uses the ground truth object motion as input; and (iii) Lin-OMOMO, which incorporates a linear interpolation strategy to generate object motion trajectories and ensure consistent object rotation throughout the sequence.

## 4.1 Qualitative Results

We compare HOI-Dyn with the SOTA HOI model CHOIS from two complementary perspectives in a synchronized manner: (i) **Action–Interaction–Response**, which evaluates the physical and semantic plausibility of object motion before and after human actions; and (ii) **Sequence-level Alignment**, which assesses the consistency of the generated HOI sequence with the input conditions and its overall causal coherence. Together, these perspectives capture both the micro-level physical causality and macro-level semantic consistency.

As shown in Figs. 2 (a) and (b), CHOIS, which lacks interaction dynamics modeling, often causes premature object motion. The objects tend to move toward expected contact points (e.g., the hand) spontaneously before human actions begin, leading to implausible behaviors such as bouncing, wobbling, or sliding. In contrast, HOI-Dyn reduces such artifacts, promoting more natural interactions. Fig. 2 (c) further highlights the difference in object responses after contact: HOI-Dyn generates subtle, physically plausible reactions (with slight sliding after a gentle kick), while CHOIS often produces exaggerated effects like flying or floating. Fig. 2 (d) shows the keyframes for sequence-level comparison. Both methods generally respect input constraints, demonstrating the effectiveness of motion diffusion. However, CHOIS often exhibits disjoint human-object motion and weak causality, while HOI-Dyn ensures consistent object motion and maintains contact, resulting in more coherent and plausible HOI sequences. More visualization results can be found in AppendixD.

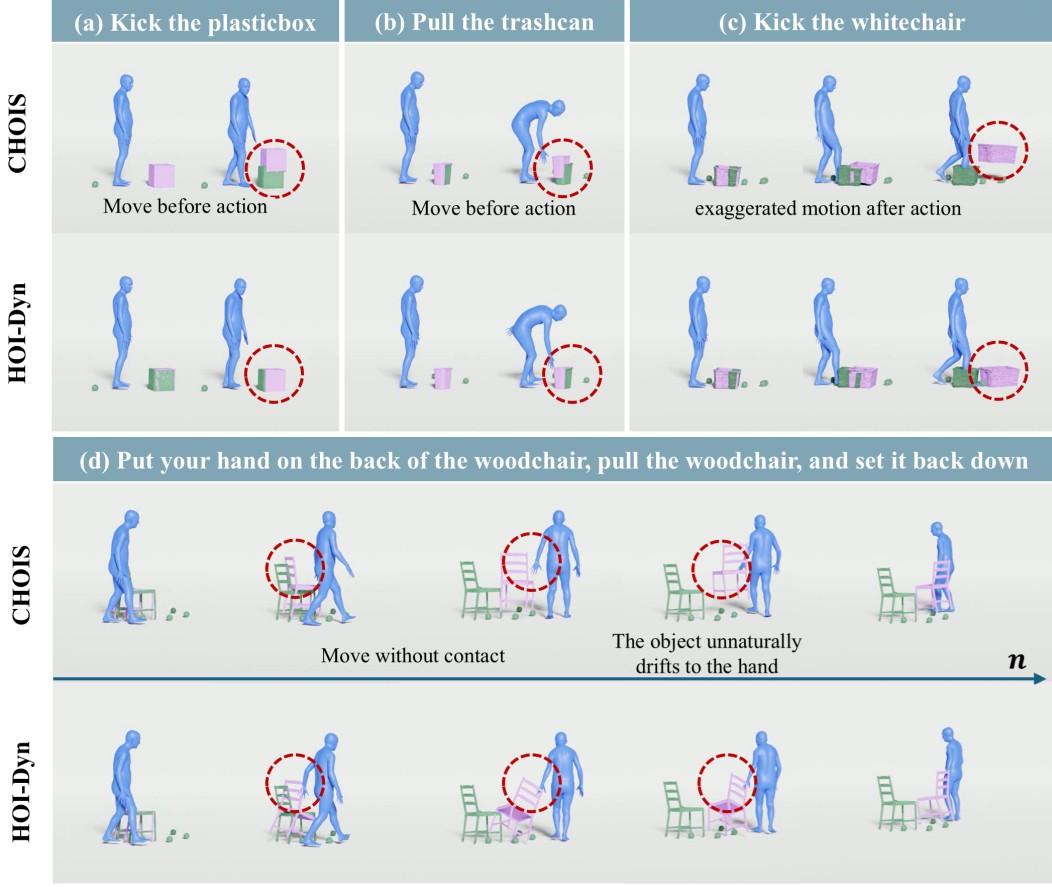

Figure 2: Comparison of HOI-Dyn and CHOIS on physical plausibility and sequence-level coherence. (a–b) CHOIS produces premature object motion lacking causal timing; (c) HOI-Dyn generates more realistic post-contact responses; (d) HOI-Dyn maintains consistent human-object interaction across the full sequence. Green markers indicate object initial state and sparse waypoints.

## 4.2 Quantitative Results

As shown in Table 1, HOI-Dyn consistently outperforms existing methods across all evaluation metrics. It achieves the lowest $T_s$ and $T_e$ in condition matching, indicating better alignment with input constraints. For human motion, it yields the best foot sliding score (FS) and FID, demonstrating a more realistic motion synthesis. In terms of interaction quality, HOI-Dyn attains the highest $C_{F1}$ and C%, while maintaining a comparable hand penetration score $P_{hand}$. It also achieves the lowest MPJPE, root translation $T_{root}$, and object motion errors ($T_{obj}, R_{obj}$), demonstrating improved accuracy in modeling human-object dynamics.

Table 1: Comparison of methods across different metrics. Arrows indicate whether lower ($\downarrow$) or higher ($\uparrow$) is better, and the same notation applies hereafter.

| Method | Condition Matching | | | Human Motion | | | Interaction | | | GT Difference | | | |
|---|---|---|---|---|---|---|---|---|---|---|---|---|---|
| | $T_s \downarrow$ | $T_e \downarrow$ | $T_{xy} \downarrow$ | $H_{feet} \downarrow$ | FS $\downarrow$ | FID $\downarrow$ | $C_{F1} \uparrow$ | C% $\uparrow$ | $P_{hand} \downarrow$ | MPJPE $\downarrow$ | $T_{root} \downarrow$ | $T_{obj} \downarrow$ | $R_{obj} \downarrow$ |
| Interdiff | 0.00 | 158.84 | 72.72 | 0.90 | 0.42 | 208.0 | 0.33 | 0.27 | 0.55 | 25.91 | 63.44 | 88.35 | 1.65 |
| MDM | 5.18 | 33.07 | 19.42 | 6.72 | 0.48 | 6.16 | 0.53 | 0.43 | 0.66 | 17.86 | 34.16 | 24.46 | 1.85 |
| Lin-OMOMO | 0.00 | 0.00 | 0.00 | 7.21 | 0.41 | 15.33 | 0.57 | 0.54 | 0.51 | 21.73 | 36.62 | 17.12 | 1.21 |
| Pred-OMOMO | 2.39 | 8.03 | 4.15 | 7.08 | 0.40 | 4.19 | 0.66 | 0.62 | 0.58 | 18.66 | 28.39 | 16.36 | 1.05 |
| GT-OMOMO | 0.00 | 0.00 | 0.00 | 7.10 | 0.41 | 5.69 | 0.67 | 0.59 | 0.55 | 15.82 | 24.75 | 0.00 | 0.00 |
| CHOIS | 2.10 | 6.16 | **3.03** | 3.39 | 0.41 | 0.87 | 0.66 | 0.54 | **0.61** | 16.01 | 24.33 | 14.29 | 0.99 |
| HOI-Dyn (Ours) | **1.75** | **5.58** | 3.26 | **3.07** | **0.37** | **0.48** | **0.71** | **0.60** | 0.64 | **15.60** | **23.90** | 12.47 | **0.90** |

To assess the generalizability of our model to novel objects, we also perform experiments on the 3D-FUTURE dataset. As shown in Table 2, although not trained on this dataset, our model significantly improves the quality of human motion and, more importantly, enhances the fidelity of human-object contact, with only a slight increase in penetration.

Table 2: Interaction synthesis results on the 3D-FUTURE dataset [30].

| Method | Condition Matching | | | Human Motion | | | Interaction | |
|---|---|---|---|---|---|---|---|---|
| | $T_s \downarrow$ | $T_e \downarrow$ | $T_{xy} \downarrow$ | $H_{\text{feet}} \downarrow$ | FS $\downarrow$ | FID $\downarrow$ | C% $\uparrow$ | $P_{\text{hand}} \downarrow$ |
| InterDiff | 0.00 | 161.26 | 72.77 | -0.26 | 0.42 | 207.3 | 0.24 | 0.11 |
| MDM | 12.58 | 40.55 | 28.72 | 7.02 | 0.49 | 8.50 | 0.34 | 0.26 |
| Lin-OMOMO | 0.00 | 0.00 | 0.00 | 6.32 | 0.42 | 23.17 | 0.44 | 0.11 |
| Pred-OMOMO | 4.15 | 9.03 | 3.89 | 6.08 | 0.40 | 3.74 | 0.50 | 0.18 |
| CHOIS | **3.23** | 6.21 | 2.99 | 2.95 | 0.42 | 1.67 | 0.47 | **0.19** |
| HOI-Dyn (Ours) | 4.60 | **6.17** | **2.95** | **2.56** | **0.37** | **1.62** | **0.54** | 0.26 |

### 4.3 Application in 3D Scene

We demonstrate the practical applicability of our method by synthesizing dynamic HOI within realistic 3D environments, conditioned on text descriptions. Specifically, we use 3D scenes from the Replica dataset [36] and manually define instructions such as *"pull the floor lamp and move it next to the sofa"*. The pipeline involves three steps: (i) parsing the textual description to identify the intended interaction and associated object(s), (ii) specifying the target object's initial placement within the scene, and (iii) planning a collision-free navigation trajectory using Habitat. The resulting path is then provided to our HOI generation model, which produces coherent and physically plausible motion sequences aligned with the instruction.

As illustrated in Fig. 3, our model successfully generates realistic agent behaviors in complex environments. In (a), the agent interacts with a floor lamp and repositions it near a sofa, while in (b), it moves a large box across the room. Both examples highlight the model's ability to generate environment-conscious and interaction-consistent motions, with potential applications in animation, virtual reality, and robotics.

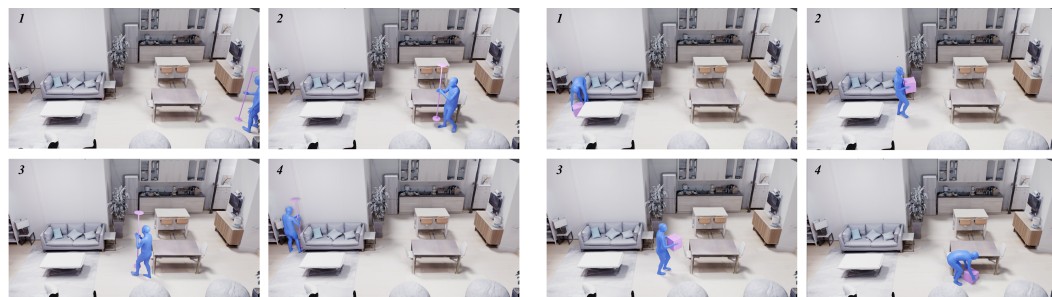

(a) Interaction with a floor lamp     (b) Interaction with a large box

Figure 3: HOI generation in realistic 3D scenes. The virtual agent interacts with different objects while maintaining physical plausibility and environmental consistency.

### 4.4 Further Discussion

**Effect of Interaction Dynamics Variants.** We analyze the effects of prediction horizon $K$, network design, and model complexity on the interaction dynamics performance. As shown in Fig. 4, a small $K$ limits large motion capture, while a large $K$ weakens subtle interaction modeling. Empirically, $K=2$ or $K=3$ yield the best results. For network design, we compare the decoupled and coupled motion strategies. Inspired by [37–39], the decoupled approach predicts object rotation and translation separately. Our coupled design instead models the unified influence of human motion and contact, yielding a better performance under similar parameter and FLOP constraints, highlighting the inherent coupling in HOI, as shown in Table 3. We also evaluate the model efficiency by varying the Transformer depth ($D$), feature dimension ($F$), and head count ($H$). The results in Table 3 show

that a lightweight model with 0.5M parameters and 0.2 GFLOPs suffices to capture high-quality interaction dynamics. For a more detailed discussion, see Appendix E

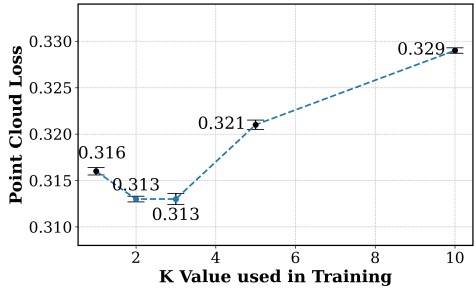

Figure 4: Effect of Horizon $K$.

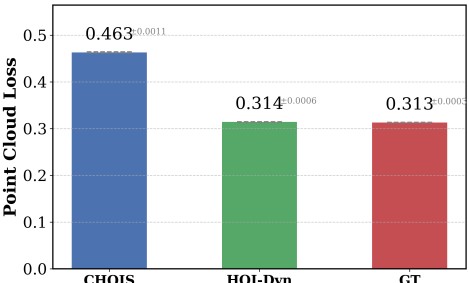

Figure 5: Object Loss via Dynamics.

Table 3: Effect of Design Variants. Our lightweight coupled model captures high-quality interaction dynamics, outperforming the decoupled variants under similar constraints.

| Architecture | Network Configuration | Object Point Cloud Loss | | | | | | Params (M) | Flops (G) |
|---|---|---|---|---|---|---|---|---|---|
| | | $K$=1 | $K$=2 | $K$=3 | $K$=4 | $K$=5 | $K=10$ | | |
| Coupled (K=1) | | $0.316^{\pm0.0004}$ | $0.514^{\pm0.0012}$ | $0.763^{\pm0.0009}$ | $1.080^{\pm0.0046}$ | $1.444^{\pm0.0051}$ | $3.422^{\pm0.0050}$ | | |
| Coupled (K=2) | | $0.313^{\pm0.0003}$ | $0.462^{\pm0.0003}$ | $0.622^{\pm0.0007}$ | $0.791^{\pm0.0024}$ | $0.982^{\pm0.0036}$ | $2.279^{\pm0.0059}$ | | |
| Coupled (K=3) | D4-F64-H8 | $0.313^{\pm0.0006}$ | $0.458^{\pm0.0016}$ | $0.603^{\pm0.0010}$ | $0.756^{\pm0.0024}$ | $0.920^{\pm0.0027}$ | $1.900^{\pm0.0079}$ | 0.483 | 0.201 |
| Coupled (K=5) | | $0.321^{\pm0.0005}$ | $0.459^{\pm0.0012}$ | $0.599^{\pm0.0013}$ | $0.736^{\pm0.0011}$ | $0.879^{\pm0.0008}$ | $1.633^{\pm0.0031}$ | | |
| Coupled (K=10) | | $0.329^{\pm0.0003}$ | $0.467^{\pm0.0022}$ | $0.604^{\pm0.0022}$ | $0.737^{\pm0.0015}$ | $0.871^{\pm0.0020}$ | $1.501^{\pm0.0009}$ | | |
| Coupled (K=2) | D1-F64-H8 | $0.349^{\pm0.0019}$ | $0.516^{\pm0.0011}$ | $0.700^{\pm0.0004}$ | $0.909^{\pm0.0023}$ | $1.144^{\pm0.0043}$ | $2.652^{\pm0.0109}$ | 0.432 | 0.057 |
| Coupled (K=2) | D4-F64-H8 | $0.313^{\pm0.0003}$ | $0.462^{\pm0.0003}$ | $0.622^{\pm0.0007}$ | $0.791^{\pm0.0024}$ | $0.982^{\pm0.0036}$ | $2.279^{\pm0.0059}$ | 0.483 | 0.201 |
| Coupled (K=2) | D8-F64-H8 | $0.318^{\pm0.0001}$ | $0.471^{\pm0.0016}$ | $0.633^{\pm0.0014}$ | $0.815^{\pm0.0052}$ | $1.035^{\pm0.0027}$ | $2.807^{\pm0.0118}$ | 0.550 | 0.394 |
| Coupled (K=2) | D8-F128-H8 | $0.563^{\pm0.0006}$ | $0.845^{\pm0.0007}$ | $1.136^{\pm0.0051}$ | $1.450^{\pm0.0052}$ | $1.791^{\pm0.0048}$ | $3.630^{\pm0.0141}$ | 0.994 | 1.552 |
| Decoupled (K=2) | (D1-F64-H8)×2 | $0.358^{\pm0.0006}$ | $0.532^{\pm0.0009}$ | $0.714^{\pm0.0027}$ | $0.919^{\pm0.0037}$ | $1.144^{\pm0.0057}$ | $2.613^{\pm0.0059}$ | 0.463 | 0.108 |
| Decoupled (K=2) | (D2-F64-H8)×2 | $0.340^{\pm0.0011}$ | $0.503^{\pm0.0006}$ | $0.676^{\pm0.0007}$ | $0.870^{\pm0.0015}$ | $1.080^{\pm0.0071}$ | $2.537^{\pm0.0049}$ | 0.496 | 0.200 |
| Decoupled (K=2) | (D4-F64-H8)×2 | $0.337^{\pm0.0006}$ | $0.503^{\pm0.0004}$ | $0.676^{\pm0.0034}$ | $0.866^{\pm0.0022}$ | $1.062^{\pm0.0047}$ | $2.409^{\pm0.0049}$ | 0.564 | 0.385 |

**Effect of Different Guidance.** Classifier-based guidance is often employed during inference in generative models. Following [13], we apply two types of guidance: *feet-floor* and *hand-object*. The *feet-floor* term encourages physical plausibility during locomotion and standing still by penalizing unnatural foot height above the ground, while the *hand-object* term enforces physically consistent contact and temporal coherence for hand–object interactions. For a more detailed description, see Appendix F. As shown in Table 4, even without guidance, our method surpasses CHOIS, especially in the interaction quality. The *feet-floor* term enhances foot realism and reduces FID, while the *hand-object* term boosts contact accuracy but slightly harms motion quality. Combining both yields consistent gains across all metrics, achieving a new SOTA.

Table 4: Effect of Different Guidance. Our method outperforms CHOIS even without guidance. The feet-floor term improves physical realism, while the hand-object term enhances contact accuracy. Combining both achieves SOTA performance across all metrics.

| Method | Condition Matching | | | Human Motion | | | Interaction | | | GT Difference | | | |
|---|---|---|---|---|---|---|---|---|---|---|---|---|---|
| | $T_s \downarrow$ | $T_e \downarrow$ | $T_{xy} \downarrow$ | $H_{\text{feet}} \downarrow$ | FS $\downarrow$ | F ID $\downarrow$ | $C_{F1} \uparrow$ | C% $\uparrow$ | $P_{\text{hand}} \downarrow$ | MPJPE $\downarrow$ | $T_{\text{root}} \downarrow$ | $T_{\text{obj}} \downarrow$ | $O_{\text{obj}} \downarrow$ |
| CHOIS (w/o gui) | 1.86 | 5.79 | 3.05 | 3.39 | 0.57 | 3.63 | 0.54 | 0.42 | 0.61 | 16.05 | 25.34 | 12.36 | 0.99 |
| CHOIS (w/ gui) | 2.10 | 6.16 | 3.03 | 3.39 | 0.41 | 0.87 | 0.66 | 0.54 | 0.61 | 16.01 | 24.33 | 14.29 | 0.99 |
| HOI-Dyn (w/o gui) | 1.56 | 5.72 | 3.03 | 5.56 | 0.37 | 3.49 | 0.60 | 0.47 | 0.61 | 15.56 | 24.61 | 11.67 | 0.91 |
| HOI-Dyn (feet-floor gui) | 1.57 | 5.56 | 3.03 | 3.23 | 0.37 | 0.66 | 0.60 | 0.46 | 0.62 | 15.48 | 23.87 | 11.28 | 0.89 |
| HOI-Dyn (hand-obj gui) | 1.77 | 5.74 | 3.27 | 5.21 | 0.37 | 2.55 | 0.71 | 0.60 | 0.64 | 15.73 | 24.56 | 13.03 | 0.91 |
| HOI-Dyn (w/ gui) | 1.75 | 5.58 | 3.26 | 3.07 | 0.37 | 0.48 | 0.71 | 0.60 | 0.64 | 15.60 | 23.90 | 12.47 | 0.90 |

**Effect of Removing Object Waypoints.** To evaluate the intrinsic capability of our interaction dynamics model, we consider a *without waypoint (w/o WP)* setting, removing predefined object waypoints for both the Baseline (CHOIS) and HOI-Dyn models. In this setting, the GT difference metric is no longer meaningful, so we focus on metrics reflecting physical plausibility and interactive realism: foot sliding, penetration (any unrealistic interpenetration involving human, object, or floor), contact F1 and accuracy, FID, and diversity. Classifier-based guidance is not applied during inference,

ensuring that results reflect the generator's learned dynamics. As shown in Table 5, HOI-Dyn demonstrates improved foot and contact quality, better FID and diversity, and comparable penetration, highlighting the reasoning and generalizability of the model.

Table 5: Performance under the without waypoint setting. HOI-Dyn improves foot and contact quality, FID, and diversity, while preserving penetration, showing its intrinsic interaction dynamics.

| Method | FS $\downarrow$ | Penetration $\downarrow$ | $C_{F1} \uparrow$ | $C_{\%} \uparrow$ | FID $\downarrow$ | Diversity $\uparrow$ |
|---|---|---|---|---|---|---|
| CHOIS w/o WP | 0.401 | **0.581** | 0.573 | 0.648 | 5.36 | 7.90 |
| HOI-Dyn w/o WP | **0.376** | 0.582 | **0.592** | **0.670** | **4.81** | **8.09** |

**Dynamics as Causality Metric.** We evaluate the interaction dynamics modeling as a proxy for causal consistency. Specifically, we compute the point cloud loss $\mathbb{E}_t[\Phi(\Delta\hat{o}_t^*, \Delta\hat{o}_t)]$ in (5), where $\Delta\hat{o}_t$ is the object's observed motion and $\Delta\hat{o}_t^*$ is the predicted response from the dynamics model. As shown in Fig. 5, despite some noise, HOI-Dyn closely aligns with the ground truth, while CHOIS shows a notable deviation, indicating a better causal modeling by HOI-Dyn. Furthermore, we visualize the frame-wise dynamics loss in Fig. 6, where the loss spikes when objects behave unrealistically. The results show that the loss remains low for HOI-Dyn, indicating that our model more effectively captures physically consistent interactions. These findings support the use of dynamics-based losses as a metric for causal evaluation in HOI synthesis.

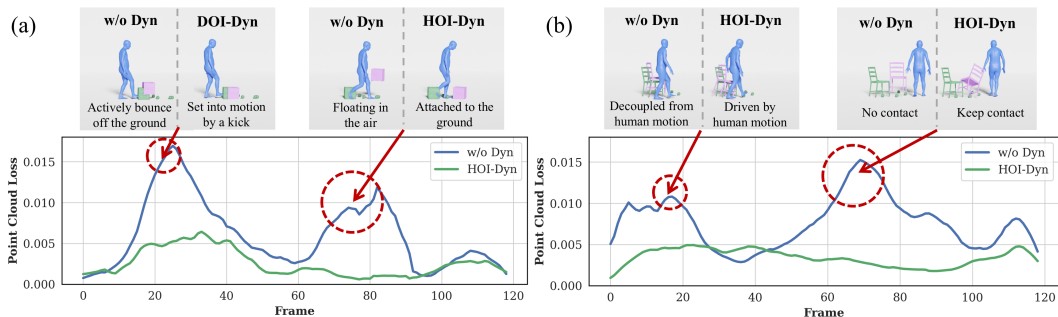

Figure 6: Frame-wise object point cloud loss. Our method HOI-Dyn (green line), consistently achieves lower loss than the baseline without dynamics supervision (blue line). Notably, high loss peaks correspond to physically implausible HOI cases, which are effectively identified by our dynamics model.

## 5   Conclusion and Limitation

In this work, we emphasize the role of interaction dynamics in ensuring physical and causal consistency in HOI generation. As an early exploration, we propose a novel driver-responder framework that explicitly models these dynamics and integrates with existing HOI motion diffusion techniques to achieve more realistic human-object interactions. While alternative approaches based on physics simulators might seem capable of modeling object dynamics more directly [40], their black-box nature, lack of differentiability, and strong reliance on precise physical properties make them difficult to incorporate into generative frameworks. A detailed discussion is provided in Appendix C.4.

Our framework currently assumes rigid objects and relies on the SMPL-X human model, which provides only coarse hand representations. Consequently, minor inaccuracies may appear in hand-object interactions, particularly in rotations (see Appendix E.3). Looking ahead, we aim to incorporate richer object attributes and enhance human modeling to better capture fine-grained interactions [41]. We also plan to explore tighter integration of HOI dependencies into generative models to further improve interaction realism [42], and extend our framework to handle multi-human and multi-object scenarios to assess scalability and practical applicability.

## Acknowledgments and Disclosure of Funding

Lin Wu was supported by a Graduate School PhD Scholarship from the College of Science and Engineering, University of Glasgow. Jianglin Lan was supported by a Leverhulme Trust Early Career Fellowship (Award No. ECF-2021-517). Zhixiang Chen acknowledges support from the N8 Research Partnership and the EPSRC (Grant No. EP/T022167/1), which provided access to the N8 Centre of Excellence in Computationally Intensive Research (N8 CIR). We also acknowledge financial support from the Division of Autonomous Systems and Connectivity within the James Watt School of Engineering, University of Glasgow.

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

## A  Linking HOI Dynamics to Classical Synchronization

### A.1  Classical Synchronization in Control Theory

To motivate our approach, we revisit the classical driver-responder (master-slave) synchronization framework from control theory, which serves as the theoretical foundation for modeling interaction dynamics in HOI generation:

$$\textbf{Driver}: \begin{cases} \dot{x}_m = F(x_m) \\ y_m = Cx_m \end{cases}, \tag{13}$$

$$\textbf{Responder}: \begin{cases} \dot{x}_s = F(x_s) + Bu_s \\ y_s = Cx_s \end{cases}, \tag{14}$$

where $x_m, x_s \in \mathbb{R}^n$ are the state vectors of the Driver and Responder systems, respectively, $y_i \in \mathbb{R}^p$ ($i = m, s$) are the outputs, $F(\cdot)$ is a Lipschitz continuous nonlinear function, $B \in \mathbb{R}^{n \times r}$ is the constant input matrix, and $C \in \mathbb{R}^{p \times n}$ is the output matrix. The control input $u_s \in \mathbb{R}^r$ is given by

$$u_s = K_s(y_m - y_s) = K_sC(x_m - x_s), \quad K \in \mathbb{R}^{r \times n}, \tag{15}$$

where $K$ is the coupling strength. The synchronization objective is defined as

$$\lim_{t \to \infty} \|x_m(t) - x_s(t)\| = 0. \tag{16}$$

This framework illustrates how a Responder can track the Driver through an error-correcting feedback signal, providing a formal basis for causal coordination.

### A.2  Interaction Dynamics Loss as Implicit Error Feedback

In HOI generation, human motion serves as the Driver, while object motion is the Responder. To enforce causal and physically plausible object responses, we introduce an **Interaction Dynamics Loss** $\mathcal{L}_{\text{dyn}}$, which measures discrepancies between generated object motion and ground-truth trajectories using a shared dynamics model $\mathcal{D}$. Conceptually, this loss functions as an error-feedback signal guiding the Responder's state toward alignment with the Driver.

During training, the diffusion model iteratively updates object states at each denoising step to minimize this error, implicitly internalizing an error-feedback control mechanism that captures causal human-object dependencies. At inference, even without access to ground-truth object motion, the model generates synchronized and physically consistent object responses by leveraging the learned driver-responder coordination. This establishes a principled connection between classical synchronization theory and modern generative HOI modeling.

## B  Details of HOI Motion and Condition Representation

**Human Motion.**  To represent the human motion $H$, we employ the SMPL-X parametric model, which reconstructs the 3D human mesh from pose and shape parameters. The pose is represented as a 204-dimensional vector, consisting of the following components: (i) a 3-dimensional translation vector that defines the joint position of the body, contributing a total of $24 \times 3$ dimensions, and (ii) the rotations of 22 body joints, contributing $22 \times 6$ dimensions to the pose vector. In addition to the pose parameters, the model includes a shape parameter vector $\boldsymbol{\beta} \in \mathbb{R}^{16}$, which controls the individual body shape.

**Object Motion.**  Object motion $O$ is characterized by two main components: the global 3D position and the relative rotation. The global position is represented by the centroid of the object, while the relative rotation, denoted as $\mathcal{R}_{\text{rel}}(t)$, describes the rotation at frame $t$ relative to the object's geometry $P$. The object vertices at frame $t$ are given by $P_t = \mathcal{R}_{\text{rel}}(t)P$. Therefore, the object motion is described by $O \in \mathbb{R}^{T \times 12}$, where each frame consists of a translation relative to the centroid and a corresponding relative rotation.

**Condition.** In our condition representation, we integrate contextual cues following the CHOIS framework. The text embedding is extracted using a pretrained CLIP text encoder. The object geometry is encoded via a MLP applied to its Basis Point Set (BPS), and the resulting features are broadcast to all frames. These BPS features are then concatenated with the initial states and waypoints (provided every 30 frames, where only the object's $x$- and $y$-coordinates are available, with the $z$-axis omitted) and further combined with the text embedding to form the condition $\mathbf{c}$. In addition, due to the fact that we include a 4-dimensional binary contact indicator that specifies hand and feet contact states in HOI data representation, we therefore construct a masked motion representation to represent the initial states and waypoint constraints $\mathbf{m} \in \mathbb{R}^{T \times (12+204+4)}$, where the initial state includes both the human pose and object pose at the first frame.

## C  Theoretical and Empirical Justification of Interaction Dynamics Loss

### C.1  Theoretical Justification via Residual Dynamics

**Definition.** We define a learned dynamics model $\mathcal{D}$ that predicts the object's relative motion based on the current scene and human motion:

$$\Delta o^*_{t \to t+k} \approx \mathcal{D}(s^{(t)}, o^{(t)}, \Delta h_{t \to t+k}; \theta_{\mathcal{D}}), \tag{17}$$

where $s^{(t)}$ is the scene context at time $t$, $o^{(t)}$ is the object state (e.g., pose), $\Delta h_{t \to t+k}$ is the future human motion, and $\theta_{\mathcal{D}}$ contains the model parameters.

To promote consistent physical behavior by the generated trajectories, we introduce an auxiliary loss that compares the residual dynamics behavior of generated and ground-truth sequences, as follows:

$$\mathcal{L}_{\text{dyn}} = \|\delta_{\mathcal{D},\text{gen}} - \delta_{\mathcal{D},\text{gt}}\|^2 \tag{18}$$

with the residuals defined as

$$
\begin{aligned}
\delta_{\mathcal{D},\text{gen}} &= \Phi\left(\mathcal{D}(\hat{s}^{(t)}, \hat{o}^{(t)}, \Delta \hat{h}_{t \to t+k}), \Delta \hat{o}_{t \to t+k}\right), \\
\delta_{\mathcal{D},\text{gt}} &= \Phi\left(\mathcal{D}(s^{(t)}, o^{(t)}, \Delta h_{t \to t+k}), \Delta o_{t \to t+k}\right),
\end{aligned}
\tag{19}
$$

where $\Phi(\cdot, \cdot)$ denotes a point-wise distance metric (e.g., the $\ell_1$ distance between the point clouds decoded from object poses).

**Bias Cancellation.** Although $\mathcal{D}$ may have approximation error, we show that comparing residuals (rather than raw predictions) reduces the effect of model bias. For theoretical clarity, we analyze in the prediction space without applying $\Phi$, and only consider $\Phi$ as a Lipschitz continuous final metric layer.

Let the true (unknown) dynamics function be $\mathcal{F}$, and define the model bias as

$$b(x) := \mathcal{D}(x) - \mathcal{F}(x). \tag{20}$$

Let $x_{\text{gen}} = (\hat{s}^{(t)}, \hat{o}^{(t)}, \Delta \hat{h}_{t \to t+k})$ and $x_{\text{gt}} = (s^{(t)}, o^{(t)}, \Delta h_{t \to t+k})$. Then the prediction-space residuals can be expressed as

$$
\begin{aligned}
\delta_{\mathcal{D},\text{gen}} &= \mathcal{D}(x_{\text{gen}}) - \Delta \hat{o}_{t \to t+k}, \\
\delta_{\mathcal{D},\text{gt}} &= \mathcal{D}(x_{\text{gt}}) - \Delta o_{t \to t+k}.
\end{aligned}
\tag{21}
$$

Subsequently, the residual difference becomes

$$
\begin{aligned}
\delta_{\mathcal{D},\text{gen}} - \delta_{\mathcal{D},\text{gt}} &= (\mathcal{D}(x_{\text{gen}}) - \mathcal{D}(x_{\text{gt}})) + (\Delta o_{t \to t+k} - \Delta \hat{o}_{t \to t+k}) \\
&= [b(x_{\text{gen}}) - b(x_{\text{gt}})] + [\mathcal{F}(x_{\text{gen}}) - \mathcal{F}(x_{\text{gt}})] + (\Delta o - \Delta \hat{o}).
\end{aligned}
\tag{22}
$$

Assuming that both $\mathcal{F}$ and $b$ are Lipschitz continuous with the Lipscthiz constants $L_{\mathcal{F}}$ and $L_b$, and that $\|x_{\text{gen}} - x_{\text{gt}}\| \leq \epsilon$, $\|\Delta \hat{o} - \Delta o\| \leq \delta$, then we obtain

$$\|\delta_{\mathcal{D},\text{gen}} - \delta_{\mathcal{D},\text{gt}}\| \leq L_b \epsilon + L_{\mathcal{F}} \epsilon + \delta. \tag{23}$$

This implies that

$$\|\delta_{\mathcal{D},\text{gen}} - \delta_{\mathcal{D},\text{gt}}\| \to 0 \quad \text{as} \quad \epsilon, \delta \to 0, \tag{24}$$

meaning the residual discrepancy vanishes if the input conditions and object motion predictions are sufficiently close. Thus, the residual-based loss is less sensitive to the absolute accuracy of $\mathcal{D}$, and instead emphasizes consistency in behavior under the same model.

**Connection to Training Loss.** While our analysis is carried out in the model output space, the actual training loss is applied using the decoded point cloud metric given by

$$\Phi(x, y) = \| f_{\text{pc}}(x) - f_{\text{pc}}(y) \|_1 \,, \tag{25}$$

where $f_{\text{pc}}(\cdot)$ denotes a deterministic decoder from object pose to point cloud. Since both $f_{\text{pc}}(\cdot)$ and $\|\cdot\|_1$ are Lipschitz continuous, the residual error behavior remains consistent under this transformation.

## C.2 Empirical Evidence from Motion Decomposition

**Setup.** To validate the approximation $\mathbb{E}_t[\delta_{\mathcal{D},\text{gen}} - \delta_{\mathcal{D},\text{gt}}] \approx 0$, we compute the residual difference

$$\Delta^{(n)}(t) = \delta_{\mathcal{D},\text{gen}}^{(n)}(t) - \delta_{\mathcal{D},\text{gt}}^{(n)}(t) \tag{26}$$

for each sequence $n \in \{1, 2, \ldots, N\}$ and time step $t \in \{1, 2, \ldots, T\}$. We then compute the aggregate statistics over all $NT$ samples, as follows:

$$\bar{\Delta} = \frac{1}{NT} \sum_{n=1}^{N} \sum_{t=1}^{T} \Delta^{(n)}(t), \quad \text{var}(\Delta) = \frac{1}{NT} \sum_{n=1}^{N} \sum_{t=1}^{T} \left( \Delta^{(n)}(t) - \bar{\Delta} \right)^2 \tag{27}$$

These metrics reflect the average residual discrepancy and its variability across all sequences and frames.

Table 6: Comparison of Statistical Measures.

| Statistic | Mean | Variance |
|---|---|---|
| Ground Truth (gt) | 0.3130 | 0.1190 |
| Generated (gen) | 0.3147 | 0.0950 |
| Difference (gen - gt) | **0.0016** | **0.1561** |

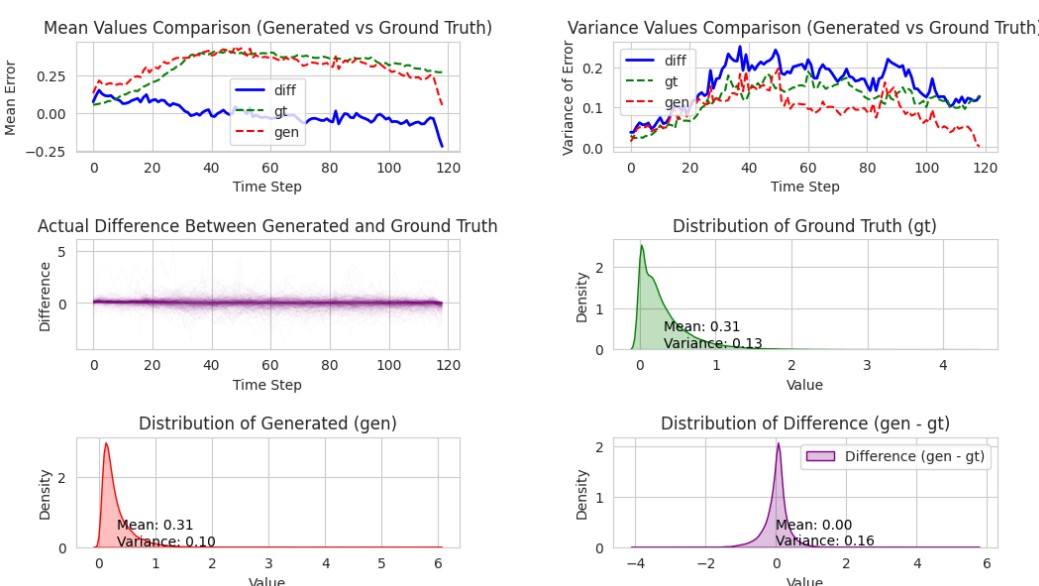

Figure 7: Statistical and distributional comparison of interaction dynamics errors for generated and ground-truth object motion. The mean and variance of residuals are nearly identical for gen and gt, while their difference is centered near zero with low variance. Probability density plots further confirm the alignment of gen and gt distributions, and support the theoretical claim of modeling bias cancellation.

**Results.** As shown in Table 6, the mean of the residual difference is nearly zero (0.0016), confirming that the residual signal has no significant bias across time steps or sequences, consistent with our theoretical approximation $\mathbb{E}_t[\delta_{\mathcal{D},\text{gen}} - \delta_{\mathcal{D},\text{gt}}] \approx 0$.

We further visualize the typical temporal patterns in Fig. 7, where the residual differences fluctuate randomly around zero, with no observable drift or systematic deviation. These empirical observations validate the core assumption that residual-based comparison cancels out the model bias when $\mathcal{D}$ is applied consistently to both generated and ground-truth trajectories.

Together with our theoretical insights, these results support the following: (i) The residual-based loss introduces **natural bias cancellation**, making the learning robust to approximation errors in $\mathcal{D}$. (ii) This enables the use of $\mathcal{D}$ as an auxiliary evaluator without requiring high-fidelity absolute predictions. (iii) The formulation enforces consistency in human-object dynamics, thereby improving the realism and coherence of generated motion. This bias-invariant design is a key strength of our auxiliary loss and contributes to the stability, generalizability, and interpretability of the learned policy.

### C.3 Residual-based Loss and Generated HOI Quality

To further evaluate the impact of our residual-based interaction dynamics loss on generated HOI quality, we conducted ablation studies comparing different formulations: (i) *Without Interaction Dynamics Loss*, i.e., baseline CHOIS without any dynamics supervision. (ii) *Residual Dynamics Loss*, i.e, our proposed formulation (10) that cancels noise and imperfections in the pretrained dynamics model. (iii) *Non-Residual Dynamics Loss*, which directly penalizes the difference between the predicted object motion $\Delta\hat{o}_t$ and the dynamics model output $\Delta\hat{o}_t^*$ without bias compensation:

$$\mathcal{L}_{\text{dyn}}^{\text{non-res}} = \mathbb{E}_t\left[\left\|\Phi\big(\Delta\hat{o}_t^*, \Delta\hat{o}_t\big)\right\|_1\right]. \tag{28}$$

We report results without classifier-based guidance to isolate pure generation capability. As summarized in Table 7, incorporating the interaction dynamics loss consistently improves generation quality over the baseline. In particular, the residual-based formulation produces more stable, physically realistic, and contact-accurate HOIs, whereas the non-residual variant allows errors in the dynamics model to propagate, potentially degrading motion quality. These findings demonstrate that the residual formulation is essential for robust HOI generation and support the design choice of our auxiliary interaction dynamics loss.

Table 7: Ablation of interaction dynamics loss formulations. Residual dynamics consistently improves physical realism and contact accuracy.

| Method | $T_{xy}\downarrow$ | FS $\downarrow$ | $C_{F1}\uparrow$ | $P_{\text{hand}}\downarrow$ | MPJPE $\downarrow$ | $T_{\text{root}}\downarrow$ | $T_{\text{obj}}\downarrow$ | $O_{\text{obj}}\downarrow$ |
|---|---|---|---|---|---|---|---|---|
| Without Interaction Dynamics | 3.05 | 3.63 | 0.54 | **0.61** | 16.05 | 25.34 | 12.36 | 0.99 |
| HOI-Dyn (Non-Residual Dynamics) | 3.03 | 0.39 | 0.59 | 0.71 | 15.63 | 25.10 | 11.86 | **0.89** |
| **HOI-Dyn (Residual Dynamics)** | **3.03** | **0.37** | **0.60** | **0.61** | **15.56** | **24.61** | **11.67** | 0.91 |

### C.4 Learned Object Motion Reactor as a Differentiable Surrogate for Physics Simulators

Physics-based approaches are widely adopted to ensure physically plausible HOI. Representative methods such as InterMimic [43], PhysHOI [40], and SkillMimic [44] rely on black-box physics simulators in combination with reinforcement learning, typically learning one policy per skill. These methods primarily focus on controlling human motion and do not explicitly model the object's reactive behavior. Moreover, their *non-differentiable nature* and dependence on detailed physical parameters (e.g., mass, friction, compliance) limit applicability in fully differentiable, end-to-end generative frameworks, particularly for long-horizon or complex interactions.

To overcome these limitations, we introduce a *learned object motion reactor*, a *differentiable, data-driven surrogate* for object dynamics. Conceptually, it functions as a **conditional world model**, producing *temporally coherent and physically plausible* behavior without enforcing explicit physical laws. Its advantages can be summarized in three key aspects:

**(i) Seamless integration with generative pipelines:** The reactor's differentiable design allows direct embedding into diffusion-based HOI generation frameworks, enabling end-to-end training and gradient-based optimization.

**(ii) Generalization without requiring physical parameters:** Unlike physics simulators that need detailed mass, friction, and compliance values, the reactor learns directly from motion patterns in data, making it robust to missing or uncertain physical parameters.

**(iii) Learning to respect physical constraints:** Although learned, the reactor captures essential behaviors such as responding appropriately to human motion and remaining stationary when not contacted, ensuring physically plausible and temporally consistent object dynamics.

By explicitly incorporating contact states, our approach further strengthens human-object coupling, extending prior work such as CHOIS [13]. Hybrid paradigms combining learned dynamics with physics-informed priors or simulators remain a promising direction; incorporating measurable physical properties, as explored in FORCE [45], could enhance realism while preserving differentiability. Nevertheless, the learned object motion reactor provides a practical, scalable, and fully differentiable solution for generative HOI modeling, effectively bridging the gap between physically grounded simulation and data-driven generation.

## D    Additional Qualitative Results of HOI Generation

We provide additional qualitative comparisons to further evaluate the effects of modeling interaction dynamics.

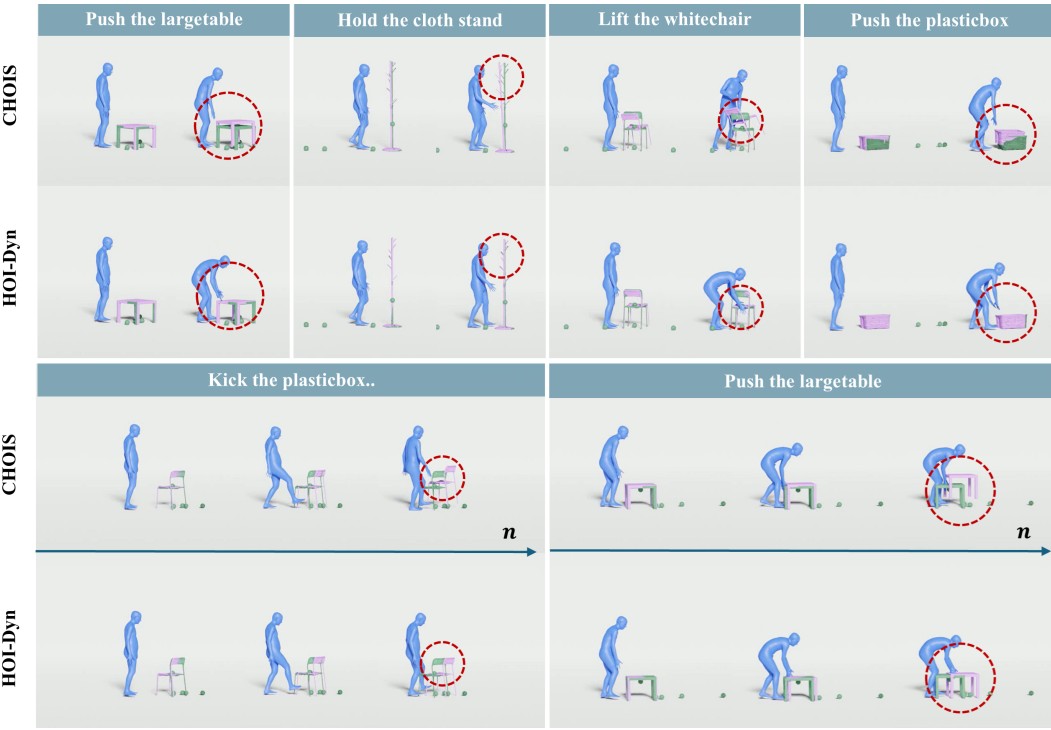

Figure 8: Qualitative comparison in Action–Interaction–Response.

In Fig. 8, we assess the *Action–Interaction–Response* loop. The results show that incorporating interaction dynamics constraints enables the model to better understand how objects should respond to both human actions and the contact context. This suggests that modeling such dynamics improves physical plausibility and contextual coherence in human–object interactions.

In Fig. 9, we further investigate *sequence-level alignment*. We observe that explicitly enforcing interaction dynamics not only enhances the consistency of human–object contact but also improves the overall motion quality. These findings align with our claim:

*"Contact is implicitly governed by the dynamics—no need to explicitly model it. If there is no contact, there is no response; if contact occurs, the object's response is naturally determined by the interaction dynamics."*

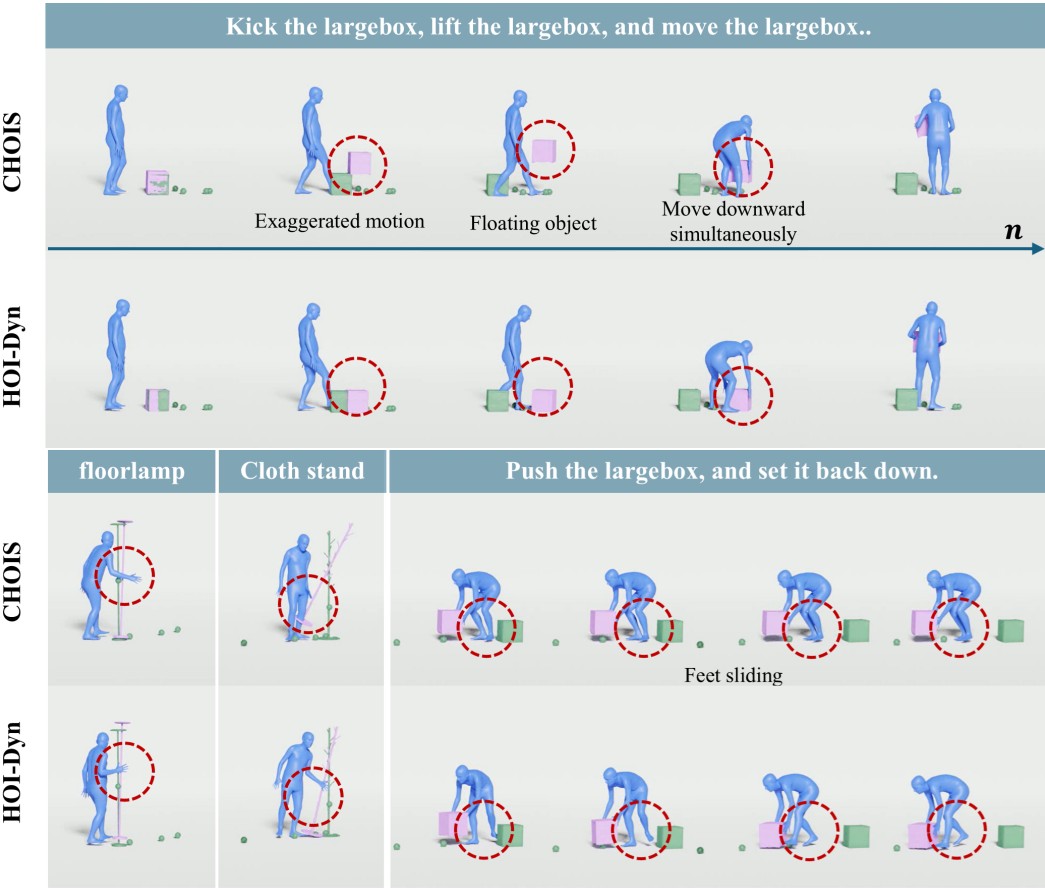

Figure 9: Qualitative comparison in Sequence-level alignment.

Moreover, by restricting the object from moving in physically implausible ways, our method encourages the human to take more active and reasonable actions. This results in motion that is not only more coordinated but also better aligned with the intended conditions or text prompts—addressing a key limitation in prior works where human and object often move independently without dynamic coherence.

## E    Network and Additional Results of Interaction Dynamics

### E.1    Model Architecture

Our dynamics model $\mathcal{D}$ is a lightweight predictor designed to infer object motion induced by human actions and interaction context. It takes as input the object-centric BPS point cloud, human joint relative motions, and contact information, and outputs the relative object motion (rotation + translation). The architecture consists of three main components:

**(i) BPS Encoder.** A two-layer multilayer perceptron (MLP) encodes the input object BPS ($1024 \times 3$) into a compact feature vector. This captures the object's shape and spatial context.

**(ii) Human Motion Module.** A Mini Transformer models the temporal interaction dynamics between the human and the object. It first encodes the joint-level motion features (3D relative positions and rotations) through a MLP, then injects condition information—concatenation of object motion, contact state, and BPS feature—via conditional encoding. A shallow transformer with learnable positional embeddings is applied, followed by **attention pooling** to obtain a global human motion feature.

**(iii) Motion Decoder.** A MLP maps the fused interaction feature to the target object motion vector $(3 + 9$ dimensions), representing translation and flattened rotation. The output can be optionally projected to $SO(3)$ to ensure a valid rotation.

## E.2 Local Smoothness and Temporal Homogeneity

Building on the design of our dynamics network, we examine how its architecture and formulation inherently enforces local smoothness and temporal homogeneity.

### E.2.1 Temporal Homogeneity

Temporal homogeneity means that the mapping from human motion to object motion is invariant to absolute time indices. Since our dynamics network is designed for step-wise interaction modeling, it only concerns the immediate effect of human motion on object motion, analogous to how applying a certain force induces a proportional response regardless of when it is applied. Intuitively, pushing with the same strength should produce comparable object displacement whether it occurs early or late in an episode. This property is directly regularized through our loss function:

$$\mathcal{L} = \mathbb{E}_{t,\, k \sim \mathcal{U}(1,K)} \left[ \frac{1}{k} \cdot \Phi(\Delta o_{t \to t+k}, \Delta o^*_{t \to t+k}) \right], \tag{29}$$

where both the starting step $t$ and horizon $k$ are uniformly sampled. The normalization factor $\frac{1}{k}$ decouples error from horizon length, ensuring that supervision is distributed evenly across different positions and scales. As a result, the model learns event-agnostic dynamics, capturing only the causal effect of human motion on object motion without bias toward specific temporal phases.

### E.2.2 Local Smoothness

**Theoretical Motivation.** Object motions in human-object interactions evolve continuously under smooth forces and contacts. Since the input human joint motion $h^{(t)}$ and object state $o^{(t)}$ are temporally continuous, the predicted object motion

$$\Delta o^{(t)} \approx \mathcal{D}(s^{(t)}, o^{(t)}, \Delta h^{(t)}; \theta_{\mathcal{D}}) \tag{30}$$

is expected to vary smoothly, forming the theoretical basis for local continuity.

**Network Design Guarantees.** The architecture, which includes differentiable MLPs, Mini-Transformer, and attention pooling, is inherently smooth. No discontinuous operations (e.g. quantization, and hard thresholds) are introduced. Training on continuous trajectories further promotes smooth transitions in the learned mapping.

**Lipschitz Continuity and Empirical Test.** We formalize local smoothness via Lipschitz continuity: a function $f$ is Lipschitz continuous with a constant $L$ if $\|f(x_1) - f(x_2)\| \le L\|x_1 - x_2\|$ for all $x_1, x_2$. To test this empirically, we perturb human motion inputs by adding random unit-vector noise scaled by $\epsilon$:

$$\tilde{\Delta h}^{(t)} = \Delta h^{(t)} + \epsilon \cdot \frac{\eta}{\|\eta\|_2}, \quad \eta \sim \mathcal{N}(0, I). \tag{31}$$

We then measure the output deviation defined as

$$\Delta \mathcal{D} = \|\mathcal{D}(s^{(t)}, o^{(t)}, \tilde{\Delta h}^{(t)}) - \mathcal{D}(s^{(t)}, o^{(t)}, \Delta h^{(t)})\|_1. \tag{32}$$

If $\Delta \mathcal{D}$ grows approximately linearly with the perturbation scale $\epsilon$, the model exhibits locally Lipschitz behavior. Tables 8–9 summarize the results, confirming smooth and proportional responses across perturbation magnitudes.

## E.3 One-Step and Auto-Regressive Prediction Results

We evaluate the predictive ability of our interaction dynamics model from two complementary perspectives: **one-step prediction** and **auto-regressive prediction**.

In the *one-step* setting, the model is given the current scene context and ground-truth human motion to predict the object state at the next frame ($T = 1$). This setting directly evaluates the local dynamics

Table 8: Effect of unit-vector perturbations scaled by $\epsilon$ on predicted object motion.

| Perturbation $\epsilon$ | Difference Norm $\Delta \mathcal{D}$ | | |
|---|---|---|---|
| | Combined (12D) | Translation (3D) | Rotation (9D) |
| $1 \times 10^{-5}$ | 2.6e-5 | 5.7e-6 | 2.5e-5 |
| $1 \times 10^{-4}$ | 1.2e-4 | 2.4e-5 | 1.1e-4 |
| $1 \times 10^{-3}$ | 1.1e-3 | 1.97e-4 | 1.07e-3 |
| $1 \times 10^{-2}$ | 1.1e-2 | 1.99e-3 | 1.09e-2 |
| $1 \times 10^{-1}$ | 1.0e-1 | 1.96e-2 | 1.00e-1 |

Table 9: Relative magnitude of input perturbations (as percentage of the average input norm). Perturbation scales correspond to $\epsilon = 10^{-5}, 10^{-4}, 10^{-3}, 10^{-2}, 10^{-1}$.

| Component | Average Norm | Perturbation Levels (%) |
|---|---|---|
| Translation (3D) | $4.42 \times 10^{-4}$ | 2.3, 23, 230, 2300, 23000 |
| Rotation (9D) | $2.83 \times 10^{-3}$ | 0.35, 3.5, 35, 348, 3480 |
| Combined (12D) | $2.87 \times 10^{-3}$ | 0.35, 3.5, 35, 348, 3480 |

response. As shown in Fig. 10, our model achieves high accuracy in object translation, with very small deviations from ground truth. However, the rotation prediction still shows room for improvement. We attribute this to the dataset lacking hand pose details, which limits the inference of fine-grained object rotations from body joints alone. Nevertheless, the predicted rotations remain broadly consistent with the ground truth.

In the *auto-regressive* setting, we recursively feed the predicted object state back into the model to generate long-term future motion. This setting reveals how prediction errors accumulate. As illustrated in Fig. 11, even though our model is only trained with one- and two-step supervision, it demonstrates impressive generalization in the auto-regressive regime. The predicted object translations stay closely aligned with the ground truth over time. Although rotation errors do increase due to compounding uncertainty, they remain stable and plausible.

Importantly, our method relies solely on the one-step supervision during training, which **avoids the inefficiency and instability of full auto-regressive rollout training**. This design enables fast training while already delivering strong performance. The model's ability to generalize well in the auto-regressive case is a *bonus*, not a requirement. Hence, the proposed design achieves an excellent balance between efficiency and long-horizon performance — a highly cost-effective formulation.

### E.4 Comparison on Interaction Dynamics

We further analyze the quality of interaction dynamics generated by CHOIS and our proposed HOI-Dyn. As shown in Fig. 12 and Fig. 13, we employ the predicted human-object sequences and compare their induced object motion against the reference dynamics. Here, the reference is not the ground truth in the test set , but the object's relative motion in the synthetic sequences used as initial conditions for prediction. Since motion synthesis is inherently multimodal, we focus on whether the predicted motion is physically and contextually plausible.

We observe that HOI-Dyn produces significantly more consistent and coherent object motion. In contrast, CHOIS often yields noisy or unstable object transitions, with noticeable discontinuities or implausible drifts over time.This comparison highlights the advantage of our synchronized control formulation and the dynamics-guided residual loss.

### E.5 Joint-Level Attention Visualization

In our proposed driver-responder synchronized control framework, we posit that once contact is established, the object's motion should predominantly follow the human motion. Based on the SMPL representation, human motion is determined by the transitions and rotations of body joints. We thus interpret object motion as a result of the aggregated influence of all human joints.

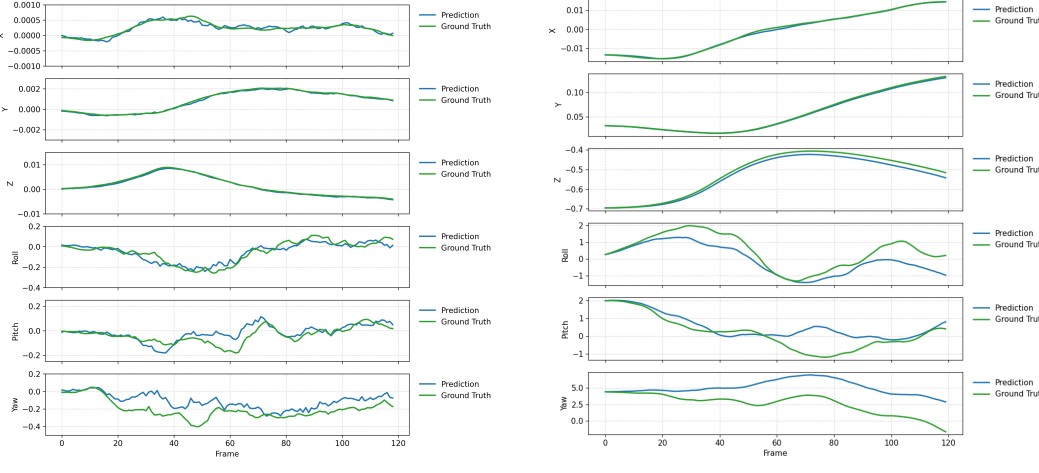

Figure 10: One-step performance.      Figure 11: Auto-regressive performance.

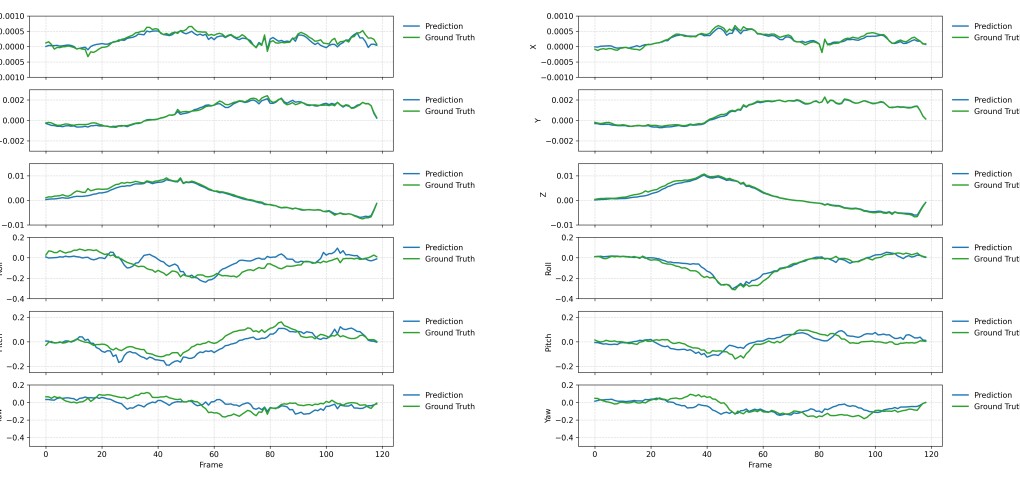

Figure 12: CHOIS.            Figure 13: HOI-Dyn.

Our `MiniTransformer` module explicitly models joint-wise contributions to object dynamics by applying attention over all $J = 24$ SMPL joints. Specifically, given joint features $\mathbf{x}_j \in \mathbb{R}^d$ for joint $j \in \{1, \ldots, J\}$, and a learned attention weight $w_j$, we compute the global representation as:

$$F_{\text{global}} = \sum_{j=1}^{J} w_j \cdot \mathbf{x}_j, \tag{33}$$

where the attention weights $w_j \in [0, 1]$ are computed via a softmax layer and satisfy $\sum_j w_j = 1$. This formulation allows the model to selectively focus on the most relevant joints under different interaction conditions, enabling interpretable and context-aware reasoning of human-object dynamics.

We visualize the joint-level attention maps for typical cases in Fig. 14 and Fig. 15. For the first example in Fig. 14, the person bends down to pick up a box. During this full-contact phase, the object is entirely under human control. The attention map highlights not only the hands but also the feet, elbows, and chest — suggesting a whole-body coordination that is both intuitive and desirable. This supports the idea of a "global response" where object motion is governed by distributed joint influence.

For another example in Fig. 15, where the person is walking while carrying the box, the attention is more concentrated on the hands and feet. This makes intuitive sense: the hands are directly manipulating the object (contributing to both translation and rotation), while the feet govern the global transition through locomotion. Across multiple sequences, we also observe consistent attention

on the hips and lower limbs, suggesting their role as stable references for interpreting whole-body movement.

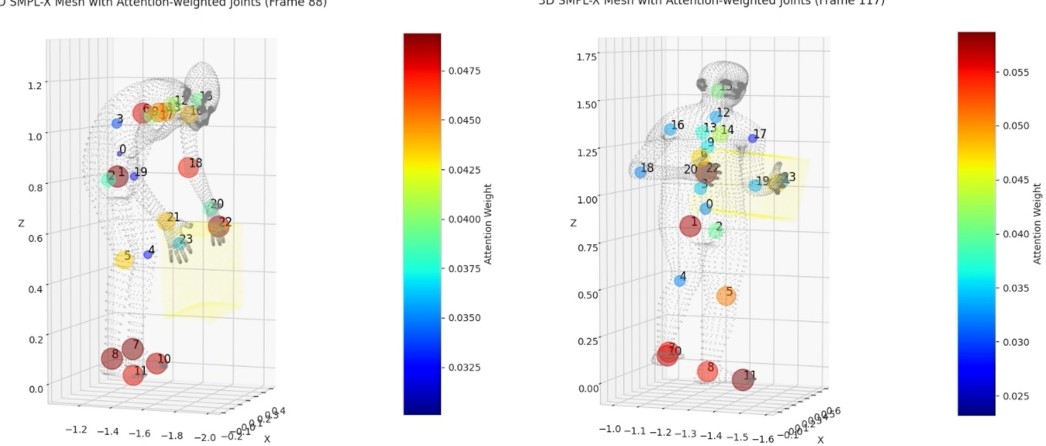

Figure 14: Pick up a box.  Figure 15: Walk forward while holding the box.

# F  Details of Classifier-based Guidance

While the use of guidance modules (e.g., contact-based constraints) can lead to targeted improvements in motion quality, all comparisons with state-of-the-art baselines are conducted under identical guidance settings. This ensures a fair evaluation, as our proposed method HOI-Dyn focuses on internally controlling and optimizing the interaction dynamics during the generation process. Therefore, the guidance configuration is kept fully consistent across all methods and does not introduce any bias in favor of HOI-Dyn.

## F.1  Feet-Floor Contact Guidance

To encourage physical plausibility during locomotion and standing still, we introduce a contact guidance loss that penalizes unnatural foot height above the ground, consistent with the CHOIS setting. Specifically, we extract the 3D positions of the left and right toe joints, and compute the support foot height as:

$$h_{\text{support}} = \min\left(h_{\text{left-toe}},\ h_{\text{right-toe}}\right), \tag{34}$$

where $h \in \mathbb{R}^{B \times T \times 1}$ represents the height (Z-axis) across batch $B$ and time $T$.

We enforce that the support foot remains near the ground $\alpha$ (set at 2 cm) via an MSE loss:

$$\mathcal{L}_{\text{feet-floor}} = \frac{1}{BT} \sum_{b=1}^{B} \sum_{t=1}^{T} \left(h_{\text{support}}^{(b,t)} - \alpha\right)^2. \tag{35}$$

This loss improves contact realism by promoting plausible foot-ground interaction.

## F.2  Hand-Object Contact Guidance

To enhance the realism and stability of hand-object interactions, we use the multi-term auxiliary loss that encourages physically plausible contact and temporal coherence.

**Contact Loss:** For each timestep, we extract the 3D positions of the left and right palm joints and compute their closest distances to the object mesh as follows:

$$\mathcal{L}_{\text{contact}} = \frac{1}{BT} \sum_{b=1}^{B} \sum_{t=1}^{T} \sum_{h \in \{\text{L,R}\}} \left[ \max(\|\mathbf{p}_h^{(b,t)} - \mathcal{M}^{(b,t)}\| - \delta, 0) \cdot \phi_{\text{contact}}^{(b,t,h)} \right], \tag{36}$$

where $\mathcal{M}^{(b,t)}$ is the object mesh at frame $t$, $\mathbf{p}_h^{(b,t)}$ is the palm position, $\delta = 0.02$ is the contact threshold, and $\phi_{\text{contact}}$ is the predicted binary contact indicator.

**Temporal Consistency Loss:** For contact frames, we enforce that the palm positions (in the object coordinate frame) remain temporally smooth and aligned by

$$\mathcal{L}_{\text{consistency}} = 2 - \frac{1}{T^2} \sum_{i,j} \left[ \cos(\theta_{i,j}^{\text{L}}) \cdot M_{i,j}^{\text{L}} + \cos(\theta_{i,j}^{\text{R}}) \cdot M_{i,j}^{\text{R}} \right], \tag{37}$$

where $\theta_{i,j}$ is the angle between the relative palm directions at frames $i$ and $j$, and $M_{i,j}^{\text{L/R}}$ are symmetric masks marking the frames where contact is active.

**Floor Penetration Penalty:** We suppress object vertices penetrating below the ground plane ($z < 0$) through

$$\mathcal{L}_{\text{floor}} = \frac{1}{N} \sum_{n=1}^{N} \max(-z_n, 0). \tag{38}$$

The final guidance loss is given by

$$\mathcal{L}_{\text{interaction}} = \mathcal{L}_{\text{contact}} + \mathcal{L}_{\text{consistency}} + \lambda \cdot \mathcal{L}_{\text{floor}}, \quad \text{with} \quad \lambda = 100. \tag{39}$$

This auxiliary loss encourages accurate and stable contact modeling.

