# OpenReview forum: "HOI-Dyn: Learning Interaction Dynamics for Human-Object Motion Diffusion"
_NeurIPS.cc/2025/Conference — NeurIPS 2025 poster_

### Official Review · Reviewer_XvKb · 2025-06-23

**Clarity:** 2
**Significance:** 3
**Originality:** 3
**Rating:** 4
**Confidence:** 4

**Summary:**

This paper presents a novel framework that models 3D human-object interactions as a driver-responder system using a lightweight transformer-based dynamics model to predict object reactions to human motions, combined with a residual-based dynamics loss for consistency.

**Questions:**

1. Unclear writing, especially regarding the workflow in Figure 1(b), which is inadequately described in the paper. Can you explain more about "causality alignment"?

2. What is error feedback in L125?

**Ethical Concerns:**

["NO or VERY MINOR ethics concerns only"]

**Final Justification:**

After reading authors' rebuttal, I think most of my concerns have been solved.  The key concern of me is why we need a learned object reactor. Based on the advantages listed by the authors, I am inclined to accept this paper.

**Limitations:**

Weakness #2 should be disscussed.

**Paper Formatting Concerns:**

There is no major formatting issues.

**Quality:**

3

**Strengths And Weaknesses:**

#### Strength
1. One notable advantage is the innovative concept of modeling humans as "drivers" and objects as "responders," which proves superior to simultaneous generation of human and object motions. This driver-responder framework better captures the causal dependency in human-object interactions, enabling more logically consistent and physically plausible motion generation by explicitly defining how human actions drive object reactions.

2. Quantitative comparisons demonstrate that the proposed method outperforms previous baselines, with significant improvements observed in key metrics with statistical significance, which validates the effectiveness and superiority of the approach.

#### Weakness

1. The absence of an ablation study on the impact of using the interaction dynamics loss during training.

2. Given that physics-based approaches are widely adopted for capturing physical plausibility in human-object interactions, the paper should give comparison or discussion on using physics simulators to model object reactions to human actions.

3. In the demonstration video, noticeable foot sliding artifacts are observed. Additionally, there are instances where the object exhibits premature movement before human contact—for example, the cases starting at 3s and 11s.

---

> ### Author Rebuttal · Authors · 2025-07-30
>
> # Response to Reviewer XvKb
>
> ## **Weakness 1**: "The absence of an ablation study on the impact of using the interaction dynamics loss during training."
> >
> >Thank you for the valuable suggestion. We had conducted ablation studies to evaluate the effectiveness of our proposed interaction dynamics loss during training. As shown in Table 4 of the submission, our method outperforms the one that does not incorporate an interaction dynamics loss.
> >
> >To further isolate the impact of the **residual formulation** in our dynamics loss (Eq. (10)), we compared it against a **non-residual variant**, which directly penalizes the difference between the predicted object motion $\Delta \hat{o}_t$ and the dynamics model output $\Delta \hat{o}_t^*$ without compensating for model bias:
> >
> > $$\mathcal{L}_{\text{dyn-non-residual}} = \mathbb{E}_t [ | \Phi (\Delta \hat{o}_t^*, \Delta \hat{o}_t ) | ]$$
> >
> >We report the results without applying classifier-based guidance to evaluate the pure generation capability, as follows:
> >
> >| Method | $T_{xy} \downarrow$ | Foot Sliding $\downarrow$ | $C_{F1} \uparrow$ | $P_{\text{hand}} \downarrow$ | MPJPE $\downarrow$ | $T_{\text{root}} \downarrow$ | $T_{\text{obj}} \downarrow$ | $O_{\text{obj}} \downarrow$ |
> >| :---- | :---- | :---- | :---- | :---- | :---- | :---- | :---- | :---- |
> >| Without Interaction Dynamics | 3.05 | 3.63 | 0.54 | 0.61 | 16.05 | 25.34 | 12.36 | 0.99 |
> >| HOI-Dyn (Non-Residual Dynamics) | 3.03 | 0.39 | 0.59 | 0.71 | 15.63 | 25.10 | 11.86 | 0.89 |
> >| **HOI-Dyn (Residual Dynamics)** | 3.03 | 0.37 | 0.60 | 0.61 | 15.56 | 24.61 | 11.67 | 0.91 |
> >
> >In summary, incorporating the interaction dynamics loss clearly improves performance over not using it. Among the various formulations, the residual version is designed to cancel noise and imperfections in the pretrained dynamics model, making it more effective (see Appendix C). The non-residual loss directly penalizes discrepancies, causing errors in the dynamics model to propagate into diffusion generation and potentially degrade performance.
> >
> >Our experiments confirm that **the residual formulation yields more stable and better results**, especially in physical realism and contact accuracy. We hope this clarifies the benefits and necessity of our interaction dynamics loss design.
> >
> ## **Weakness 2**: "Given that physics-based approaches are widely adopted for capturing physical plausibility in human-object interactions, the paper should give comparison or discussion on using physics simulators to model object reactions to human actions."
> >
> >Thank you for the insightful suggestion. Different from the purpose of this paper, most existing physics-based methods—such as Intermimic [1], PhysHOI [2], and SkillMimic [3]—typically **rely on black-box physics simulators and reinforcement learning to train one policy per skill**, focusing on human control rather than object dynamics. As a result, they **do not explicitly model the object’s reactive behavior in response to human motion**, and are rarely applicable to HOI generation within a fully differentiable, end-to-end trainable pipeline.
> >
> >Although our approach follows a kinematics-based paradigm for HOI generation, we have made considerable efforts to incorporate physical principles.
> >
> >Our method **introduces a learned dynamics module that explicitly predicts the object’s motion** conditioned on human pose, contact, and object state. This can be viewed as a conditional world model, allowing our generator to capture temporally coherent and physically consistent reaction patterns, while retaining **full differentiability for efficient gradient-based optimization**.  Moreover, our method follows the approach of CHOIS by explicitly **introducing contact states** to enhance human-object coupling in the diffusion process, and further extends it by **modeling the object’s motion response to human actions**. This enables improved physical reasoning and more accurate generation.
> >
> >We acknowledge the promising direction of integrating learning-based approaches with physics simulators in future work. In particular, incorporating physical properties such as mass, friction, and compliance—as explored in datasets like FORCE [5]—could further enhance realism and fidelity. We will clarify these distinctions and include a dedicated discussion comparing our approach with physics-based methods in the revised version.
> >
> >**References:**
> >
> >[1] InterMimic: Towards Universal Whole-Body Control for Physics-Based Human-Object Interactions. CVPR 2025.
> >[2] PhysHOI: Physics-Based Imitation of Dynamic Human-Object Interaction. arXiv 2023.
> >[3] SkillMimic: Learning Basketball Interaction Skills from Demonstrations. arXiv 2024.
> >[4] CHOIS: Controllable human-object interaction synthesis. ECCV 2024\.
> >[5] FORCE: Dataset and Method for Intuitive Physics Guided Human-object Interaction. 3DV 2025.
> >
> ## **Weakness 3**: "In the demonstration video, noticeable foot sliding artifacts are observed. Additionally, there are instances where the object exhibits premature movement before human contact—for example, the cases starting at 3s and 11s."
> >
> >Thank you for your valuable observation. Compared to the baselines, our method has already substantially reduced object drift through the interaction dynamics loss, as reflected in improved contact quality and lower motion errors (Table 1). While minor drift remains in a few frames—likely due to the soft nature of our constraint—overall, our method achieves higher Contact F1 (0.71 vs. 0.66), lower translation (12.47 vs. 14.29) and rotation errors (0.90 vs. 0.99), and improved human motion quality (Foot Sliding 0.37 vs. 0.41, Human FID 0.48 vs. 0.87), suggesting more consistent interactions and more natural motions.
> >
> ## **Question 1**: "Unclear writing, especially regarding the workflow in Figure 1(b), which is inadequately described in the paper. Can you explain more about 'causality alignment'?"
> >
> >Thank you for the question. **Causality alignment** refers to ensuring that the **object motion is a plausible and physically consistent response to the human motion and interaction context**.
> >
> >While diffusion models can generate realistic sequences, object motion may still be inconsistent with how objects should physically react. To address this, we introduce an **interaction dynamics model** $\mathcal{D}$, which learns how objects typically respond to human motion and context.
> >
> >During training, we use $\mathcal{D}$ to simulate object motion from both the generated and ground-truth HOI sequences. We then compare the simulated outcomes to measure whether the generated motion leads to realistic object responses. This forms our **Interaction Dynamics Loss (Eq. (10))**, which enforces **causal consistency** between human actions and object reactions.
> >
> >In short, Figure 1(b) shows this process: **using $\mathcal{D}$ as a learned judge to align the causal effect of human-object interactions**. We will revise the figure caption and related text in the manuscript for clarity.
> >
> ## **Question 2**: "What is error feedback in L125?"
> >
> >Thank you for pointing this out. We appreciate the opportunity to clarify the role of the error feedback $e(y_h^{(t)}, y_o^{(t)})$ in our framework.
> >
> >In our formulation, we decouple the human and object dynamics through a **driver-responder** structure: the human acts under internally guided dynamics (as is natural in human motion), while the object motion is externally driven and must respond to physical interaction cues. This asymmetry is reflected in the control signal $u^{(t)}$, which governs the object's behavior in response to the human.
> >
> >The **error feedback** $e(y_h^{(t)}, y_o^{(t)})$ is a conceptual signal that reflects the mismatch between the human’s intent and the object’s actual behavior. In the context of training, this feedback takes the form of the **Interaction Dynamics Loss (Eq. (10))**, i.e., the difference between the generated object motion and the ground-truth motion via the dynamics model $D$.
> >
> >Although $u^{(t)}$ is not explicitly parameterized as a separate control module, it is **implicitly optimized through the diffusion process**. During training, each denoising step updates the object’s predicted state to better match the ground truth. This is achieved by minimizing the diffusion model’s reconstruction loss, which, in effect, serves as a surrogate for the accumulated error feedback across time steps. Thus, we interpret the internal optimization of the diffusion model as a form of **control signal learning**—the model is learning to steer object motion in a way that aligns with the human motion and interaction context.
> >
> >At inference time, although no ground-truth object motion is available, the model has learned to generate synchronized and physically plausible object behaviors by internalizing this driver-responder coordination during training. In this way, the **error feedback drives the learning of the interaction dynamics**, and the **diffusion model acts as an implicit controller**.

---

> > ### Comment · Reviewer_XvKb · 2025-08-01
> >
> > Thank you for your detailed rebuttal addressing the concerns raised. I think weakness #1 and questions #1 and #2 have been adequately addressed.
> >
> > With respect to Weakness #2, I believe that a physics simulator itself serves as a robust object reaction predictor. Moreover, it inherently ensures that objects remain stationary when not in contact with humans - consistent with the concern of weakness #3, which has also been noted by Reviewer UuR7. So the main concern is - why we need the proposed object motion reactor?

---

> > > ### Author Response · Authors · 2025-08-01
> > >
> > > >We are happy to see that most concerns have been addressed. Thank you again for your thoughtful follow-up on Weakness #2.
> > > >
> > > >While we agree that a physics simulator can serve as a robust object reaction predictor, our motivation for introducing a learned object motion reactor stems from a fundamental difference in setting. In physics simulation, such behaviors are governed by explicit physical laws. But in **generative models without access to a simulator**, these behaviors must be learned; otherwise, the model may produce implausible motion, such as drifting or floating.
> > > >
> > > >
> > > >Our reactor addresses this by acting as a **differentiable, data-driven surrogate** that predicts object motion conditioned on human motion, object state, and interaction context. This enables:
> > > >
> > > >- **Seamless integration with generative pipelines**: Unlike simulators, our model is lightweight and fully compatible with end-to-end diffusion-based HOI generation.
> > > >
> > > >- **Generalization without requiring physical parameters**: Simulators require detailed physical inputs (e.g., mass, friction), >which are often unavailable in real-world HOI data. Our reactor learns directly from motion patterns in data.
> > > >
> > > >- **Learning to respect physical constraints**: Although learned, our model captures key behaviors—such as responding appropriately to human motion.
> > > >
> > > >In summary, while physics simulators provide physically grounded behavior, our learned reactor is essential for capturing such patterns in scalable, data-driven generative models—producing physically plausible, coherent, and more usable HOI sequences.

---

> > > > ### Comment · Reviewer_XvKb · 2025-08-04
> > > >
> > > > Thanks for the detailed and thoughtful response. Most of my concerns have been addressed, and I will raise my rating accordingly.

---

### Official Review · Reviewer_UuR7 · 2025-06-24

**Clarity:** 3
**Significance:** 3
**Originality:** 3
**Rating:** 5
**Confidence:** 4

**Summary:**

The paper proposes a transformer-based dynamics model that predicts object movements induced from human movements during interaction. The dynamics model is used to calculate dynamics loss when training a conditional diffusion-based generative model which outputs human object interaction from object geometry/text/waypoint conditional input. The paper demonstrates the dynamics-based approach improves human-object interaction quality (consistency, contact, natural and plausible human/object movements, etc).

**Questions:**

This paper introduces interaction dynamics through a clever and novel approach using dynamics models and residual-based dynamics loss. The method achieves improved HOI motion synthesis results compared to SOTA baselines, which is demonstrated through comprehensive experiments. Despite the presence of motion artifacts (Weakness 1), the work represents a valuable advancement.
However, it would be better if: (1) more clarification that the assumptions made in using the dynamics model for calculating dynamics loss (Weakness 2), and (2) action-specific baseline comparisons to more clearly demonstrate the advantages of the dynamics-based approach (Weakness 3).

**Ethical Concerns:**

["NO or VERY MINOR ethics concerns only"]

**Final Justification:**

The rebuttal addresssed most of my concerns, and therefore I would keep my original rating

**Limitations:**

yes

**Quality:**

3

**Strengths And Weaknesses:**

Strengths

1. Defining interaction dynamics and consistency as object movement induced by human movement is innovative and intuitive. The integration of dynamics loss into CHOIS's HOI learning diffusion model is clever, introducing no additional inference overhead while enabling conditional generative models to learn interaction dynamics.

2. Defining dynamics as residuals is effective and the paper demonstrates that such formulation mitigates possible errors from the dynamics model, making the approach plausible and robust.

3. The method shows improved results compared to baselines, with extensive experiments conducted across various settings for the dynamics model.

Weaknesses

1. The result videos exhibit common artifacts where objects drift when not in contact with humans (at demo video 00:02, 00:12, 00:16). A contact-aware term that prevents object moving when human-object interaction has not yet occurred would improve realism.

2. While L181 states that the dynamics model is locally smooth and temporally homogeneous, the local smoothness assumption requires more explanation. The claim that "local smoothness reflects the continuity of physical motion" needs more detail on how the dynamics model ensures continuity.

3. It would be better if performance improvement is analyzed per action type (e.g., "put," "kick") in Table 1. In the video, the CHOIS baseline appears effective for hand-based interaction (carry, push, etc) but degrades significantly for foot-based interactions compared to the paper's method, and it would be clear for readers if this difference is demonstrated numerically.

(minor weakness for better clarity)

4. Please include input text in videos so readers can evaluate whether the output motion is semantically aligned with the given text.

---

> ### Author Rebuttal · Authors · 2025-07-30
>
> # Response to Reviewer UuR7
> >
> ## **Question 1**: "The result videos exhibit common artifacts where objects drift when not in contact with humans (at demo video 00:02, 00:12, 00:16). A contact-aware term that prevents object moving when human-object interaction has not yet occurred would improve realism." (Weakness 1)
> >
> >Thank you for your valuable observation. Compared to the baselines, our method has already substantially reduced object drift through the interaction dynamics, which is reflected in improved Contact quality and lower motion errors (Table 1). Only minor drift remains in a few frames, likely due to the soft nature of our constraint. We agree that incorporating stronger contact-aware mechanisms could further enhance realism, and we plan to investigate this in future work.
> >
> ## **Question 2**: "While L181 states that the dynamics model is locally smooth and temporally homogeneous, the local smoothness assumption requires more explanation. The claim that "local smoothness reflects the continuity of physical motion" needs more detail on how the dynamics model ensures continuity." (Weakness 2)
> >
> >Thank you for raising this point. We clarify how local smoothness is ensured from three perspectives: **theoretical motivation, architectural design, and empirical validation**.
> >
> >**(1)Theoretical Motivation**: The assumption of local smoothness is rooted in the physics of interaction: object dynamics in human-object interactions follow Newtonian mechanics, where object motions evolve continuously in time in response to smooth forces and contact. Since human joint motion $h^{(t)}$ and object state $o^{(t)}$ are both temporally continuous signals in the real world, their influence on $\Delta o^{(t)}$ should also vary continuously. This forms the basis of the following formulation:
> > $$ \Delta o^{(t)} \approx \mathcal{D}(s^{(t)}, o^{(t)}, \Delta h^{(t)}; \theta\_{\mathcal{D}}), $$
> >where the learnable function $\mathcal{D}$ models the object's next-step change as a function of the current state and human-induced motion. Since the inputs vary smoothly across time, the model is expected to learn a smooth mapping unless otherwise constrained.
> >
> >**(2)Architectural Design**: We do not introduce any discontinuous operators (e.g., quantization, hard thresholding). The network is composed of standard differentiable components (MLPs, attention, etc.), which are smooth by construction. Additionally, the input space of $\mathcal{D}$ is locally dense (i.e., trained on continuous trajectories), promoting the learning of smooth transitions. Temporal homogeneity is further ensured as the model is **time-step agnostic**—i.e., it is applied uniformly across all frames without access to absolute time.
> >
> >**(3)Empirical Validation**: We validate local continuity by adding Gaussian perturbations to the input human joint states and tracking how the predicted object dynamics change. As shown below, the changes in predicted translation and rotation grow smoothly and proportionally with the perturbation scale ($\epsilon$),  consistent with the behavior of a Lipschitz-continuous system.
> >
> >| Perturbation $\epsilon$ | Combined (12D) Diff Norm | Trans (3D) Diff Norm | Rot Diff (9D) Norm |
> >| :---- | :---- | :---- | :---- |
> >| $1 \times 10^{-5}$ | 2.6e-5 | 5.7e-6 | 2.5e-5 |
> >| $1 \times 10^{-4}$ | 1.2e-4 | 2.4e-5 | 1.1e-4 |
> >| $1 \times 10^{-3}$ | 1.1e-3 | 1.97e-4 | 1.07e-3 |
> >| $1 \times 10^{-2}$ | 1.1e-2 | 1.99e-3 | 1.09e-2 |
> >| $1 \times 10^{-1}$ | 1.0e-1 | 1.96e-2 | 1.00e-1 |
> >
> >To contextualize the perturbation magnitudes, the average norms of the input joint delta motions are approximately:
> >
> >| Component | Average Norm | Perturbation Level Relative to Average Norm (%) |
> >| :---- | :---- | :---- |
> >| Translation (3D) | $4.42 \times 10^{-4}$ | 2.3%, 23%, 230%, 2300%, 23000% (for $\epsilon = 1e{-5}$ to $1e{-1}$) |
> >| Rotation (9D) | $2.83 \times 10^{-3}$ | 0.35%, 3.5%, 35%, 348%, 3480% |
> >| Combined (12D) | $2.87 \times 10^{-3}$ | 0.35%, 3.5%, 35%, 348%, 3480% |
> >
> >This shows the perturbations range from very small to several times the natural input scale, providing a meaningful test of local smoothness.
> >
> >**Summary**: The local smoothness assumption is physically grounded, structurally preserved, and empirically validated in our setup. We are happy to clarify this in the revised version by expanding the discussion in L181 and including the continuity test results.
> >
> ## **Question 3**: "It would be better if performance improvement is analyzed per action type (e.g., "put," "kick") in Table 1\. In the video, the CHOIS baseline appears effective for hand-based interaction (carry, push, etc) but degrades significantly for foot-based interactions compared to the paper's method, and it would be clear for readers if this difference is demonstrated numerically." (Weakness 3)
> >
> >We sincerely appreciate the reviewer’s insightful suggestion regarding analyzing performance improvement by action types.
> >
> >**Action category assignment:** Since each textual description often contains multiple action verbs, we adopt a **multi-label classification** scheme. If a text contains any foot-related action (e.g., “kick”), it is assigned to the Foot-Interaction category; if it contains any hand-related action (e.g., “grab”, “pull”), it is assigned to the Hand-Manipulation category.
> >
> >**Quantitative results:** We conducted detailed analysis of key metrics on these two categories (Hand-Manipulation and Foot-Interaction), summarized in the table below:
> >
> >| Metric | CHOIS-Hand | Ours-Hand | CHOIS-Foot | Ours-Foot |
> >| :---- | :---- | :---- | :---- | :---- |
> >| Foot Sliding ↓ | 4.16 | **3.69 (↓11.4%)** | **3.48** | 3.50 (↑0.6%) |
> >| Contact F1 ↑ | 0.67 | **0.72 (↑7.6%)** | 0.33 | **0.37 (↑12.1%)** |
> >| Penetration ↓ | 0.57 | **0.56 (↓1.8%)** | 1.10 | **1.03 (↓6.4%)** |
> >| MPJPE ↓ | 16.02 | **15.61 (↓2.6%)** | 13.32 | **12.94 (↓2.8%)** |
> >| $T_{\text{obj}}$ ↓ | 14.36 | **12.46 (↓13.2%)** | 13.32 | **9.80 (↓26.4%)** |
> >| $O_{\text{obj}}$ ↓ | 0.98 | **0.90 (↓8.2%)** | 0.74 | **0.72 (↓2.7%)** |
> >
> >These results show that our method outperforms CHOIS baseline on both hand- and foot-based interactions, with particularly notable improvements in foot-interaction cases where the baseline struggles more. We will incorporate this fine-grained analysis into the revised manuscript to enhance clarity and help readers better appreciate our method’s advantages.

---

> > ### Comment · Reviewer_UuR7 · 2025-08-05
> >
> > Thanks for the detailed response and quantitative results. Please include the results in the final version. I have no further questions and would like to keep my rating.

---

### Official Review · Reviewer_biUj · 2025-07-01

**Clarity:** 3
**Significance:** 3
**Originality:** 3
**Rating:** 4
**Confidence:** 5

**Summary:**

This work introduces a novel driver–responder perspective for human-object interaction (HOI) generation, where human actions drive object responses. Central to this approach is the concept of interaction dynamics, which governs how objects naturally react to human motion—implicitly handling contact and ensuring physically plausible interactions.

**Questions:**

1. One point of curiosity is how the model would perform if the object waypoint constraints were removed. Since the current framework relies on predefined waypoints to guide HOI generation, it would be insightful to evaluate whether the interaction modeling still holds up without such guidance. This could better demonstrate the model’s true capacity for reasoning about physical causality and interaction dynamics.

**Ethical Concerns:**

["NO or VERY MINOR ethics concerns only"]

**Limitations:**

yes

**Quality:**

3

**Strengths And Weaknesses:**

### **1. Strength:**
- The entire article is basically complete and the experiments are adequate.
- The overall qualitative and quantitative results are satisfactory, showing clear improvements over CHOIS across several metrics and visual examples.
- Interaction Dynamics modeling perform well on the object pose correction.

### **2. Weakness:**
- The related work section is somewhat rough and lacks precision in categorizing and positioning recent HOI generation methods. For instance, HOI-Diff and CG-HOI are concurrent works, so the statement that CG-HOI is further enhanced is inaccurate and potentially misleading. Additionally, in Section 2.3, I would not categorize CHOIS under joint human-object prediction. Similar to OMOMO, CHOIS relies heavily on predefined object trajectories, effectively simplifying the task. A more appropriate categorization would be to group methods like HOI-Diff, CG-HOI, InterDreamer [1], THOR, HIMO [2], and ChainHOI [3] together, as they focus on jointly modeling human-object interactions without access to future object motion. OMOMO, CHOIS and [4] should be categorized into Object trajectory-guided HOI generation.


[1] InterDreamer: Zero-Shot Text to 3D Dynamic Human-Object Interaction. Sirui Xu et al.

[2] HIMO: A New Benchmark for Full-Body Human Interacting with Multiple Objects. Liang Xu et al.

[3] ChainHOI: Joint-based Kinematic Chain Modeling for Human-Object Interaction Generation. Ling-An Zeng

[4] Human-Object Interaction from Human-Level Instructions. Zeng Wu et al.

---

> ### Author Rebuttal · Authors · 2025-07-30
>
> # Response to Reviewer biUj
>
> ## **Weakness 1**: "The related work section is somewhat rough and lacks precision in categorizing and positioning recent HOI generation methods"
> >
> >Thank you for the valuable feedback. We agree that HOI-Diff and CG-HOI [1] are concurrent works and will revise the phrasing accordingly. We also appreciate your suggested taxonomy separating joint modeling methods (e.g., HOI-Diff [2], CG-HOI [1]) from object trajectory-guided ones (e.g., CHOIS [3], OMOMO [4]), and will update Section 2.3 to reflect this clearer categorization.
> >
> ## **Question 1**: "One point of curiosity is how the model would perform if the object waypoint constraints were removed. Since the current framework relies on predefined waypoints to guide HOI generation, it would be insightful to evaluate whether the interaction modeling still holds up without such guidance. This could better demonstrate the model’s true capacity for reasoning about physical causality and interaction dynamics."
> >
> >We appreciate your insightful concern.Importantly, our proposed dynamics model **does not rely on using these navigation points** as input. To isolate and assess the actual contribution of the interaction dynamics model, we design a **No WayPoint (NO-WP)** setting, in which we **remove waypoints** from both the Baseline (CHOIS[3]) and the proposed HOI-Dyn models.
> >
> >Under this setting, the comparison metric *GT difference* is no longer meaningful. Therefore, we focus on those metrics that are more relevant to physical plausibility and interactive realism, including the metrics of **foot sliding, penetration, contact F1, contact accuracy, FID**, and **Diversity**. The classifier-based guidance is inapplicable during testing; thus, the reported results reflect the generator’s intrinsic performance. The results are as follows:
> >
> >| Metric | Foot Sliding ↓ | Penetration ↓ | Contact F1 ↑ | Contact Acc ↑ | FID ↓ | Diversity ↑ |
> >| :---- | :---- | :---- | :---- | :---- | :---- | :---- |
> >| Baseline NO-WP | 0.401 | 0.581 | 0.573 | 0.648 | 5.36 | 7.90 |
> >| HOI-Dyn NO-WP | **0.376** | 0.582 | **0.592** | **0.670** | **4.81** | **8.09** |
> >
> >Compared to the baseline, our HOI-Dyn (NO-WP) significantly improves foot sliding, contact F1, contact accuracy, FID, and diversity, while maintaining comparable penetration levels. These results highlight the effectiveness of the interaction dynamics.
>
> ---
>
> >**References:**
> >\[1\] Cg-hoi: Contact-guided 3d human-object interaction generation, CVPR 2024\.
> >\[2\] Hoi-diff: Text-driven synthesis of 3d human-object interactions using diffusion models, CVPRW 2025\.
> >\[3\] Controllable human-object interaction synthesis, ECCV 2024\.
> >\[4\] Object motion guided human motion synthesis, TOG 2023\.

---

> > ### Comment · Reviewer_biUj · 2025-08-04
> > **Comment by Reviewer**
> >
> > Thank you for the detailed responses. I have no further questions and will maintain my current rating.

---

### Official Review · Reviewer_J5sP · 2025-07-05

**Clarity:** 2
**Significance:** 2
**Originality:** 3
**Rating:** 4
**Confidence:** 3

**Summary:**

This paper introduces HOI-Dyn, a framework for generating more realistic 3D human-object interactions. It models interactions as a "driver-responder" system, where human actions drive object reactions. The core uses a lightweight Transformer model to predict object responses and incorporates the loss of a residual dynamic for consistency. This dynamics model is only used during training for efficient inference. Experiments show that HOI-Dyn improves HOI generation quality and provides a new evaluation metric.

**Questions:**

1. Can this modeling approach be extended to settings with multiple objects or multiple people?
2. Should the absolute position and pose of the object be considered separately? It seems their reactive behaviors would follow different patterns. If they're mixed, could this lead to overfitting?
3. What's the fundamental advantage of this method compared to prior approaches that model human, object, and interaction separately? Intuitively, this modeling approach seems more suited for a single human and a single object. In scenarios with more people and objects, this method might limit scalability and significantly increase problem complexity.

**Ethical Concerns:**

["NO or VERY MINOR ethics concerns only"]

**Final Justification:**

The authors have provided a compelling rebuttal that successfully addresses my primary concerns regarding scalability and their unified modeling approach.

The most significant point is the clear demonstration that their model's core framework is not inherently limited to single-human, single-object scenarios. Their proposed extension, which uses a permutation-invariant aggregation strategy, offers a logical and theoretically sound path to scale the model for multi-human interactions. This thoughtful generalization shows the approach's potential beyond the current, focused implementation.

Furthermore, the authors' empirical study on coupled versus decoupled modeling is a crucial addition. The results are decisive: the coupled model consistently outperforms the decoupled version. This data-driven finding is a strong justification for their design choice, as it proves that modeling an object's translation and rotation jointly provides a beneficial inductive bias. It correctly reflects how these motions are interconnected in real-world human-object interactions. The authors have convincingly shown that their model's design is both well-justified and leads to superior performance.

**Limitations:**

yes

**Quality:**

3

**Strengths And Weaknesses:**

Strengths:
1. The idea of modeling HOI generation as a reactive system is interesting and offers a novel perspective on the problem.
2. The assumption of an active-passive system is very reasonable because objects shouldn't move without contact.

Weaknesses:
1. Can this modeling approach be extended to settings with multiple objects or multiple people?
2. Should the absolute position and pose of the object be considered separately? It seems their reactive behaviors would follow different patterns. If they're mixed, could this lead to overfitting?
3. What's the fundamental advantage of this method compared to prior approaches that model human, object, and interaction separately? Intuitively, this modeling approach seems more suited for a single human and a single object. In scenarios with more people and objects, this method might limit scalability and significantly increase problem complexity.

---

> ### Author Rebuttal · Authors · 2025-07-30
>
> # Response to Reviewer J5sP
>
> ## **Question 1**: "Can this modeling approach be extended to settings with multiple objects or multiple people?"
> >
> >Thank you for the question. Yes, our modeling approach *can be extended to scenarios involving multiple humans and objects*. For detailed discussions and the formulation of such extensions, please refer to our response to Question 3 below.
>
> ## **Question 2**: "Should the absolute position and pose of the object be considered separately? It seems their reactive behaviors would follow different patterns. If they're mixed, could this lead to overfitting?"
>
> >Thank you for the insightful suggestion. Our interaction dynamics model focuses on the object’s motion state changes, predicting relative translation and rotation rather than absolute position and pose. This relative formulation better captures reactive behaviors. To investigate joint versus separate modeling of translation and rotation, we compare two design choices as discussed in Section 4.3: (1) **Coupled** modeling, where translation and rotation are jointly predicted by a shared module; and (2) **Decoupled**, where separate modules independently predict translation and rotation. To ensure a fair comparison, we fix the maximum horizon to $K=2$ and control the model size (i.e., size of the transformer module) in both design choices.
> >
> >| Transformer Config | Model Type | Params (M) | VAL Loss | TEST Loss |
> >| :---- | :---- | :---- | :---- | :---- |
> >| Depth4\_Dim64\_Head8 | Coupled | 0.483 | 0.1318 ± 0.0004 | **0.3130 ± 0.0004** |
> >| Depth8\_Dim64\_Head8 | Coupled | 0.550 | *0.1300 ± 0.0003* | 0.3177 ± 0.0001 |
> >| (Depth2\_Dim64\_Head8) $\times$ **2 branches** | Decoupled | 0.496 | 0.1455 ± 0.0013 | 0.3396 ± 0.0015 |
> >| (Depth4\_Dim64\_Head8) $\times$ **2 branches** | Decoupled | 0.564 | 0.1387 ± 0.0010 | 0.3370 ± 0.0006 |
> >| (Depth4\_Dim128\_Head8) $\times$ **2 branches** | Decoupled | 1.047 | **0.1264 ± 0.0005** | 0.3382 ± 0.0010 |
> >
> >**Key observation**:
> >
> >* Under the current model scale and training setup, *coupled models consistently demonstrate better generalization performance compared to decoupled ones*.
> >
> >Though decoupled models with larger capacity can achieve lower validation loss, their test loss remains higher, indicating a larger generalization gap. This suggests that the coupled design introduces a helpful inductive bias by jointly reasoning about translation and rotation, which often co-occur in physically realistic human-object interactions.
> >
> >In summary, the coupled approach is more parameter-efficient and it better captures the interaction dynamics under the current settings, while decoupled modeling’s benefits may appear in more complex scenarios that need separate treatment of translation and rotation.
>
> ## **Question 3**: "What's the fundamental advantage of this method compared to prior approaches that model human, object, and interaction separately? Intuitively, this modeling approach seems more suited for a single human and a single object. In scenarios with more people and objects, this method might limit scalability and significantly increase problem complexity."
> >
> >### (1) Fundamental Advantage of the Proposed Method
> >
> >Our method’s key advantage lies in integrating human and object motions within a unified generative framework, as described in Section 2.3. This framework leverages the strong capability of diffusion models to produce globally coherent sequences, which is crucial for capturing the causal flow in human-object interactions. By explicitly decoupling the driver (human) and responder (object) roles through interaction dynamics, our approach ensures that object reactions are causally consistent with human motion—**addressing common issues where interactions lack clear cause-effect relationships**.
> >
> >In contrast, approaches that model human and object separately often fail to naturally capture their interactions. Even some unified models suffer from artifacts—such as objects moving before human initiation—because **generative models often prioritize overall distributional coherence over detailed local interactions**. Our method not only harnesses the strengths of unified generative modeling but also explicitly enforces fine-grained interaction through structured dynamics.
> >
> >### (2) Scalability to More Objects and Humans
> >
> >Thank you for the valuable comment. Our current work focuses on single-human, single-object interactions, aiming to isolate and examine core interaction dynamics under a controlled setting. Since our interaction dynamics model is **object-centric**, in multi-human and multi-object scenarios, the modeling focuses on each object individually. The generalization to these scenarios can be naturally decomposed into two typical cases:
> >
> >#### Case 1: Object Being Manipulated by One Human
> >
> >This is fully covered by our current formulation. As a reminder, our single-human interaction dynamics (Eq.2) are expressed as :
> >
> >$$ \Delta o^{(t)} \leftarrow \mathcal{D}(s^{(t)}, o^{(t)}, \Delta h^{(t)}; \theta\_{\mathcal{D}}) $$
> >
> >where $o^{(t)}$ is the current object state, $\Delta h^{(t)}$ is the human motion change, $s^{(t)}$ encodes the interaction context such as contact type and object geometry.
> >
> >#### Case 2: Object Being Manipulated by Multiple Humans
> >
> >To handle scenarios where multiple humans simultaneously interact with the same object (e.g., jointly carrying a table), we propose the following generalization of Eq.2:
> >
> >$$ \Delta o^{(t)} \leftarrow \mathcal{D}(  o^{(t)},   {(s_i^{(t)}, \Delta h_i^{(t)})}_{i=1}^N  ; \theta\_{\mathcal{D}})$$
> >
> >Each pair $(s_i^{(t)}, \Delta h_i^{(t)})$ captures the interaction context and motion of the $i$-th human. This formulation enables the model to reason over heterogeneous human-object interactions — such as different contact locations, force directions, or limb configurations — in a unified manner.
> >
> >To integrate the set ${(s_i^{(t)}, \Delta h_i^{(t)})}_{i=1}^N$, we can adopt a **permutation-invariant aggregation** strategy, such as attention pooling or a transformer-based encoder, yielding a compact interaction embedding. This embedding, combined with the object state $o^{(t)}$, serves as input to the dynamics model to predict object motion under multi-human manipulation.
> >
> >We believe this generalization offers a principled foundation for scaling up interaction dynamics modeling from single-human cases to more realistic, multi-human scenarios. However, due to the limited availability of large and diverse multi-human interaction datasets, we have not yet performed extensive experiments in this setting. We will update the manuscript to include these discussions and plan to further explore experimental validation in future work.

---

> > ### Comment · Area_Chair_vkDm · 2025-08-06
> >
> > Dear Reviewer J5sP,
> >
> > Could you please read the authors' rebuttal and respond whether your concerns have been addressed or not? Thank you!
> >
> > Best regards,
> >
> > Your AC

---

> > ### Comment · Reviewer_J5sP · 2025-08-08
> >
> > Thank you for your detailed response. I will be raising my score.

---

### Decision · Program_Chairs · 2025-09-17

**Decision:**

Accept (poster)

**Comment:**

This paper received overall positive reviews after the rebuttal (3 borderline accepts and 1 accept).

The main reasons to accept the paper are summarized as follows.
- Reviewers consistently identified the core "driver-responder" formulation as a primary strength. They described the idea of modeling HOI generation as a reactive system as "interesting" (J5sP), "innovative" (UuR7, XvKb), and offering a "novel perspective" (J5sP). This framework was seen as better capturing the causal dependency in interactions (XvKb).

- The technical approach of integrating the dynamics model as a training-time loss was recognized by Reviewer UuR7, as it introduces no additional inference overhead. The use of a residual formulation for the dynamics was also noted as "effective" for mitigating potential errors from the dynamics model, making the approach plausible and robust (UuR7).

- The paper was recognized for its satisfactory quantitative and qualitative results. Reviewers noted that the experiments were "adequate" and showed "clear improvements" over the CHOIS baseline across several metrics (biUj), and that the method "shows improved results compared to baselines, with extensive experiments" (UuR7).

At the same time, there are negative concerns, including:
- The paper was criticized for a lack of precision in the related work section. One reviewer (biUj) pointed out that the categorization of prior methods was inaccurate and potentially misleading, and that some concurrent works were not positioned correctly.

- Reviewers noted the presence of visual artifacts in the generated results. One reviewer (UuR7) observed that objects sometimes drift when not in contact with the human, and another (XvKb) pointed to noticeable foot sliding and instances of objects moving prematurely before contact.

- The paper was seen as lacking a sufficient ablation study on the impact of the interaction dynamics loss during training (XvKb). Additionally, a comparison or discussion regarding the use of physics simulators as an alternative for modeling object reactions was noted as a missing component (XvKb).

The AC overall recognizes the significance of this paper on learning interaction dynamics for human-object interaction generation using diffusion models and also that the existence of visual artifacts is inevitable due the difficulty of the task. The AC recommends accepting this paper. The authors are highly recommended to fix the first and third concerns listed above and also incorporate new results in the rebuttal into the revision.